

# Target categorization of aerosol and clouds by continuous multiwavelength-polarization lidar measurements

Holger Baars[1], Patric Seifert[1], Ronny Engelmann[1], and Ulla Wandinger[1]

[1]Leibniz Institute for Tropospheric Research (TROPOS), Permoser Str. 15, 04318 Leipzig, Germany

*Correspondence to:* Holger Baars (holger.baars@tropos.de)

**Abstract.** Absolute calibrated signals at 532 and 1064 nm and the depolarization ratio from a multiwavelength lidar are used to categorize primary aerosol but also clouds in high temporal and spatial resolution. Automatically derived particle backscatter coefficient profiles in low temporal resolution (30 min) are applied to calibrate the lidar signals. From these calibrated lidar signals, new atmospheric parameters in temporally high resolution (quasi particle backscatter coefficient) are derived. By using

thresholds obtained from multi-year, multi-site EARLINET measurements, four aerosol classes (small; large, spherical; large, non-spherical; mixed, partly non-spherical) and several cloud classes (liquid, ice) are defined. Thus, particles are classified by their physical feature (shape and size) instead of a classification by source.

The methodology is applied to two months of continuous observations (24 hours, 7 days a week) with the multiwavelength-Raman-polarization lidar Polly[XT] during the HOPE campaign in spring 2013. Cloudnet equipment was operated continuously

directly next to the lidar and is used for comparison. By discussing three 24-h case studies, it is shown that the aerosol discrimination is very feasible and informative and gives a good complement to the Cloudnet target categorization. By analyzing the entire HOPE campaign, almost 1 million pixel (5 min times 30 m) could be successfully classified from the two months data set with the newly developed tool. We find that the majority of the aerosol, trapped in the planetary boundary layer (PBL), were small particles as expected for a heavily populated and industrialized area. Large, spherical aerosol was observed mostly

at the top of the PBL and close to the identified cloud bases indicating the importance of hygroscopic growth of the particles at high relative humidity. Interestingly, it is found that on several days non-spherical particles were dispersed into the atmosphere from ground.

## 1  Introduction

Aerosol and clouds are important atmospheric players influencing weather and climate. In contrast to long-lived gaseous com-

ponents in the atmosphere, these components are short-lived and feature a strong spatiotemporal variability. Aerosols act as cloud condensation nuclei and ice nucleating particles and are thus one major driver for cloud optical and microphysical properties and precipitation initiation. Because aerosol is emitted from various sources and is short-living, several aerosol types with different optical and microphysical properties exist in different heights of the atmosphere influencing solar radiation and clouds in different ways. Therefore, the climate effects of aerosol directly and of aerosols on clouds (indirectly) are still very

uncertain (IPCC, 2013).



In order to better quantify the spatiotemporal distribution of aerosol and clouds as well as to improve the determination of their interaction, it is inevitable to observe aerosol and clouds, best in 4D, but realistically round the clock and vertically resolved. Active satellite-based sensors such as CALIPSO (Cloud-Aerosol Lidar and Infrared Pathfinder Satellite Observation, Winker et al., 2009), Cloudsat (Stephens et al., 2002), CATS (Cloud-Aerosol Transport System, Yorks et al., 2016), and, planned for

future space missions, ADM(Atmospheric Dynamics Mission)-Aeolus (Stoffelen et al., 2005) and EarthCARE (Earth Clouds, Aerosols and Radiation Explorer, Illingworth et al., 2015) cover the globe but with low temporal and spatial resolution. Thus, also high-performance ground-based observations are needed. Scientific networks such as Cloudnet (Illingworth et al., 2007) or the ARM (Atmospheric Radiation Measurement) Climate Research Facilities (Mather and Voyles, 2013) use different ground-based instruments at the same location (supersite) to gather as much information as possible in high temporal and spatial (i.e.,

vertical) resolution but at specific locations only. Cloudnet for example uses as minimum instrumentation a cloud radar, a ceilometer (a simple backscatter lidar) and a microwave radiomter (MWR) to characterize the atmosphere above the supersite. Cloudnet delivers several products, ranging from calibrated measurements to microphysical cloud properties. Very well known and widely used is the Cloudnet target categorization (Hogan and O'Connor, 2004), which classifies a series of different particle types in the observed atmospheric column (e.g. liquid droplets, ice crystals, aerosols, etc.). However, Cloudnet is in its

current state not able to distinguish different aerosol types which is a prerequisite to constrain aerosol-cloud-interaction studies and to improve the continuous estimation of the radiative properties of aerosol.

Active remote sensing with lidar is a key technique for characterizing aerosols, able to capture the atmospheric state on a vertically resolved basis usually covering the whole troposphere. For an intense characterization of aerosol type and properties, so-called multiwavelength lidars (MWL) are applied using the synergistic information from different wavelengths, scattering

mechanisms and polarization state of the received light (e.g., Ansmann and Müller, 2005).

Optical aerosol properties have been widely investigated by using lidar profiles in low temporal resolution, applying the traditional Raman method (Ansmann et al., 1992), the Klett-Fernald method (Klett, 1981; Fernald, 1984), and the depolarization method (e.g., Cairo et al., 1999) to determine the intensive properties (Ångström exponents, extinction-to-backscatter (lidar) ratio, particle depolarization ratio) of different aerosol types and their mixtures (Müller et al., 2007; Tesche et al., 2011; Ans-

mann et al., 2011; Burton et al., 2012; Pappalardo et al., 2013; Groß et al., 2013; Giannakaki et al., 2015; Amiridis et al., 2015; Sugimoto et al., 2014; Illingworth et al., 2015; Baars et al., 2016).

Based on such measurements, classification schemes for aerosol from high-resolution lidar measurements have been developed for space-borne lidars (CALIPSO, Omar et al. 2009; EarthCARE, Illingworth et al. 2015; Groß et al. 2015), airborne-High Spectral Resolution Lidar (HSRL) measurements (Burton et al., 2012; Groß et al., 2013), some specific ground-based

instruments (ARM, Darwin, Australia, Thorsen et al. 2015), and lidar networks focussing on the determination of mineral dust concentration in Asia (Asian Dust Network AD-NET, Shimizu et al. 2010; Sugimoto et al. 2014).

Due to recent advances in hardware, small sophisticated ground-based MWL (e.g., Polly[XT] lidar systems, Engelmann et al., 2016), which can run unattended and autonomously 24h/7days a week, have been developed and are deployed globally. Motivated by this technical progress, we aimed at developing a stand-alone tool for continuously running multiwavelength-

polarization lidars for a basic categorization of the observed particles (targets) in analogy to the Cloudnet target categorization.



With this tool we want to obtain an estimate of the dominant type of backscatterer (molecules, aerosol types, clouds) which then can be used for further intensive studies and to complement the Cloudnet target categorization. For this approach we focus on the derivation of certain key parameters, which are not needed with high accuracy, but are sufficient to perform a first estimate of certain particle types in the atmosphere. The basic lidar quantities used are the attenuated backscatter coefficients at 532 nm and 1064 nm (calibrated range-corrected lidar signal) and the calibrated volume linear depolarization ratio at 532 nm. These key parameters are highly useful as they are available for many continuously measuring lidar systems worldwide, e.g., the lidars within PollyNET (Baars et al., 2016), AD-NET, and the spaceborne lidar CALIPSO. From these lidar paramters, further products have been developed to allow a first-guess particle typing.

To develop this tool and demonstrate the feasibility, potential, and limitations of this approach, we have used the unique data set obtained during the High-Definition Clouds and Precipitation for advancing Climate Prediction (HD(CP)$^2$) prototype experiment HOPE (Macke et al., 2016) in western Germany. The MWL Polly$^{\mathrm{XT}}$ (Engelmann et al., 2016) and the Cloudnet instruments (cloud radar, ceilometer, MWR) were deployed in the frame of the Leipzig Aerosol and Cloud Remote Observations System (LACROS, Bühl et al., 2013) next to each other at Krauthausen, Germany, continuously for two months in Spring 2013. Polly$^{\mathrm{XT}}$ is a sophisticated, compact, scientific multiwavelength lidar to which the quality-assurance procedures proposed by EARLINET (The European Aerosol Research Lidar Network, Pappalardo et al., 2013) are applied. Without such high-quality measurements, a proper aerosol characterization as described in the following is not possible. The collocation of the instruments makes the derived data set a perfect environment for developing an aerosol classification from MWL while the Cloudnet categorization can be performed in parallel.

For the HOPE data set, we perform a so-called absolute calibration on the lidar observations from automatically derived particle backscatter profiles (Baars et al., 2016) and derive temporally high resolved atmospheric parameters which allow us to estimate size and shape and finally type of the particles in the atmosphere. This basic typing can then be used for detecting different aerosol layers, for further research like on aerosol-cloud-interaction processes, or as input for calibration procedures to automatically retrieve optical properties of the observed particles (e.g., D'Amico et al. 2015) and finally even for retrieving microphysical properties (Müller et al., 2016; Veselovskii et al., 2015) which then may lead to an advanced particle categorization (e.g., HETEAC (hybrid end-to-end aerosol classification), Wandinger et al., 2016).

The paper is structured as follows: First, the HOPE campaign, i.e., location and instrumentation, is briefly introduced in Sec. 2. The methodology to derive quantitative lidar parameters with temporal high resolution is explained in Sec. 3. Next, the new target categorization is introduced and intensively discussed by means of three case study days during HOPE in Sec. 4. The new methodology was applied on the complete HOPE data set and analyzed in section 5. Finally, conclusions are drawn in Sec. 6.

## 2 HOPE

During the HD(CP)$^2$ prototype experiment HOPE (Macke et al., 2016), the multi-wavelength-Raman lidar (MWL) Polly$^{\mathrm{XT}}_{\mathrm{IfT}}$ (Al-thausen et al., 2009; Engelmann et al., 2016) was deployed at Krauthausen (50.879746°N, 6.414571°E, 110 m asl), near Jülich,



western Germany, in April and May 2013as part of the LACROS facility (Bühl et al., 2013). A detailed description of the campaign together with the prevailing meteorological conditions can be found in Macke et al. (2016).

Polly$_{\mathrm{IfT}}^{\mathrm{XT}}$ (System version labeled "IfT", cp. Engelmann et al., 2016) is an automatic, portable multi-wavelength-polarization Raman lidar with automatic calibration procedures of latest standards which was operated in 24/7 mode during HOPE. The

5 lidar emits pulses of linearly polarized light at 355, 532, and 1064 nm at a repetition frequency of 20 Hz. The receiver has 7 channels detecting the elastically backscattered light at the three aforementioned wavelengths, the cross-polarized light at 532 nm, and the vibrational Raman scattered light at 387, 407, and 607 nm. With Polly$_{\mathrm{IfT}}^{\mathrm{XT}}$, aerosol profiles can be obtained with 30 m vertical and 30 s temporal resolution. The full overlap between the laser beam and the receiver field of view is about 1500 m, so that an overlap correction (Wandinger and Ansmann, 2002) is applied below this height. The lidar was operated in

10 photon-counting mode. The system is pointed 5° off-zenith to prevent the detection of specular reflection by the planar planes of horizontally oriented ice crystals (Hu et al., 2009; Westbrook et al., 2009). A detailed description of the system including a quality assessment can be found in Engelmann et al. (2016).

Furthermore, a cloud radar, a Doppler wind lidar, a ceilometer, and an AERONET (Aerosol Robotic Network) sun photometer were deployed next to the lidar as part of the LACROS facility. From these instruments, Cloudnet products (Illingworth et al.,

15 2007) and AERONET products (Holben et al., 2001) are available. Because of radar scanning experiments during HOPE-Jülich, Cloudnet products which require vertically pointing measurements are sporadically not available for this campaign.

## 3  Methodology

In modern multiwavelength lidars a number of different receiving channels are installed to make use of as much information from the atmosphere as possible (elastic and Raman (inelastic) scattering, change in polarization state due to scattering, etc.).

20 In this way, high-quality aerosol products are obtained on a vertically resolved basis. However, because of the high background noise, Raman lidar observations during daytime are challenging. Therefore, for continuous (24/7) measurements, we concentrate on the use of channels for elastic backscattering, including depolarization. The key challenge to succeed with automated aerosol retrievals is the calibration of the lidar signals. There are two main tasks before an automated aerosol target categorization can be performed: The calibration of the backscatter profiles and the calibration of the depolarization products.

25 ### 3.1  Calibration of backscatter

The backscatter signal strength $P$ for a certain range $R$ at the wavelength $\lambda$ can be described for each channel by:

$$P^\lambda(R) = C^\lambda \frac{O^\lambda(R)}{R^2} \left[ \beta_{\mathrm{par}}^\lambda(R) + \beta_{\mathrm{mol}}^\lambda(R) \right] \exp \left\{ -2 \int_0^R \left[ \alpha_{\mathrm{par}}^\lambda(r) + \alpha_{\mathrm{mol}}^\lambda(r) \right] \mathrm{d}r \right\} \tag{1}$$

with the wavelength-dependent lidar system parameter $C^\lambda$ containing all instrument-relevant quantities, the overlap function $O^\lambda(R)$, the molecular (subscript $\mathrm{mol}$) and particle (subscript $\mathrm{par}$) backscatter coefficient $\beta$, and the atmospheric transmissivity described by the molecular and particle extinction coefficient $\alpha$. The molecular backscatter and extinction coefficients can



easily be calculated from pressure and temperature profiles obtained from radio soundings or model output with well-known scattering formulas (Bucholtz, 1995). For usual lidar applications, the particle backscatter coefficient is obtained by applying the Raman (Ansmann et al., 1992) or Klett-Fernald method (Klett, 1981; Fernald, 1984) to the received signals. With these methods, the lidar signal is calibrated in a certain height range of the atmosphere for which only molecular scattering is as-
sumed. However, these methods require appropriate weather conditions (e.g., no low-level clouds) and temporal averaging over typically at least 30 minutes to increase the signal-to-noise ratio (SNR) in the calibration height region. Thus, for temporally high-resolved 24/7 aerosol analysis, these methods are not applicable. Therefore, we perform an absolute lidar calibration by deriving the lidar system parameter $C^\lambda$ to obtain foremost the attenuated backscatter coefficient.

For the measurements performed during HOPE, $C^\lambda$ was derived from particle backscatter coefficient profiles which were au-
tomatically computed with the Raman or Klett-Fernald methods at 30-min resolution as described in Baars et al. (2016). From these profiles, $C^\lambda(R)$ can be calculated by rearranging Eq. 1 to

$$C^\lambda(R) = \frac{P^\lambda(R) R^2}{\left[\beta_{\text{par}}^\lambda(R) + \beta_{\text{mol}}^\lambda(R)\right] O^\lambda(R)} \exp\left\{2\int_0^R \left[\alpha_{\text{par}}^\lambda(r) + \alpha_{\text{mol}}^\lambda(r)\right] \mathrm{d}r\right\}. \qquad (2)$$

The final $C^\lambda$ is computed as the mean value of a height window of 1000 m above the full overlap height (i.e., 1500 m in case of $\text{Polly}_{\text{IfT}}^{\text{XT}}$) and is considered to be height-independent. For the automatically retrieved particle backscatter profiles from the
Polly systems, all known instrumental issues (e.g., background substraction) which could cause height-dependent effects were corrected except for the partial overlap in the lowermost part of the lidar profile described by $O^\lambda(R)$ which is a substantial feature of each lidar system. For that reason and because the particle extinction coefficient derived with the Raman method is only available during night time, we decided to make a hybrid approach to solve Eq. 2. First, we use a particle extinction coefficient profile derived from the particle backscatter coefficient profile multiplied with a constant lidar ratio of 55 sr as a
good compromise of the lidar ratio values observed during HOPE and at other European continental sites (clean and polluted continental aerosol, desert dust, and smoke, Mattis et al., 2004; Müller et al., 2007; Groß et al., 2013; Schwarz, 2016; Baars et al., 2016). Second, we assume height-independent extinction below 500 m to account for both, the incomplete overlap within the lidar profile and atmospheric variability in the lowermost troposphere. At 500 m, already more than 80% of the overlap between the laser and the telescope field-of-view are reached and the applied overlap correction profile can correct the signal
trustworthy up to the full overlap height.

Figure 1 shows the daily mean lidar system parameters calculated as described above for the HOPE campaign. For some days, no calculation was possible due to unfavourable weather conditions and thus the unavailability of automatically retrieved backscatter profiles for calibration. Vertical dashed lines indicate setup changes in the lidar. Even though we tried to minimize setup changes (neutral density filters, overlap adjustment, laser energy, emission-window cleaning), several changes were
necessary but not always influencing the derived lidar system parameter.

One can see that during most of the intervals of no setup change, the lidar system parameter is relatively stable and only some of the setup changes have caused a significant change in $C^\lambda$. However, there are also periods were there was a significant change





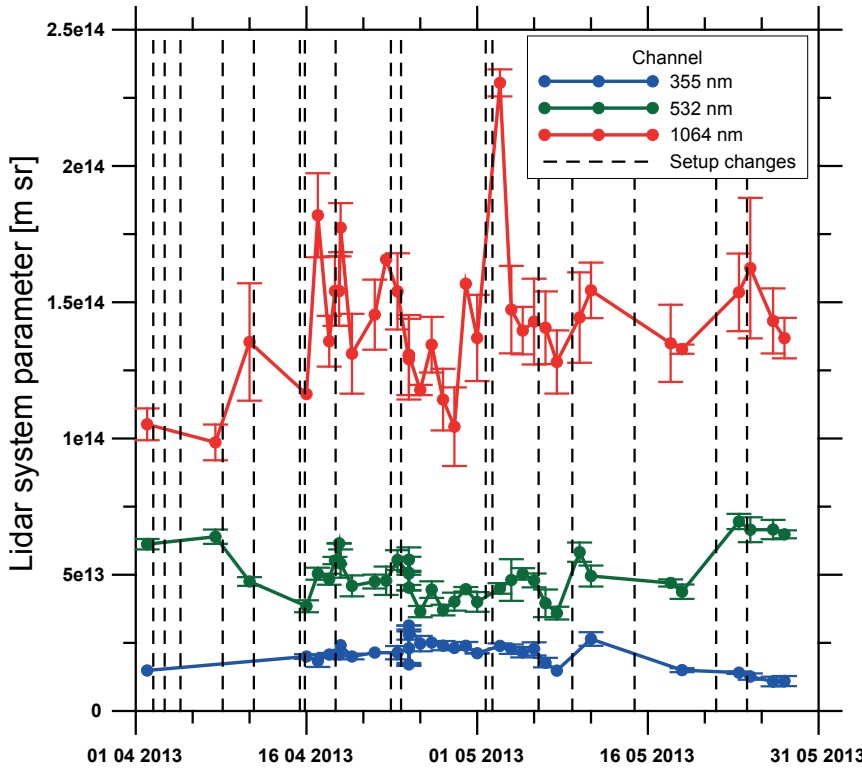

**Figure 1.** Lidar system parameter $C^\lambda$ for 355 nm, 532 nm, and 1064 nm respectively. Vertical lines indicate lidar setup changes.

of $C^\lambda$ even without changes in the setup, such as between 21 April 2013 and 01 May 2013. It was found that changes in the indoor temperature of the cabinet due air conditioning malfunctioning had led to a change of the alignment and thus a change in $C^\lambda$ during this period. On two afternoons (25 Apr and 10 May), the corresponding data were therefore not considered in the analysis. The daily mean lidar system parameter can finally be obtained with a standard deviation of less than 20%. On three days (18 April, 25 April, and 10 May), for which multiple system setup changes were performed, more than one lidar system parameter was used to account for these setup changes. In all other cases, the daily mean system parameter was used when available, otherwise the closest lidar system parameter from the days before/after was applied to calculate the calibrated attenuated backscatter coefficient derived by dividing the range-corrected signal with the lidar system parameter:

$$\beta_{\text{att}}^\lambda(R) = \frac{P^\lambda(R)R^2}{C^\lambda} = \left[\beta_{\text{par}}^\lambda(R) + \beta_{\text{mol}}^\lambda(R)\right] \exp\left\{-2\int_0^R \left[\alpha_{\text{par}}^\lambda(r) + \alpha_{\text{mol}}^\lambda(r)\right]\mathrm{d}r\right\}. \tag{3}$$

## 3.2 Calibration of depolarization ratio

The calibration of the depolarization measurements of Polly$^{\text{XT}}$ systems is done with the so-called $\Delta 90°$-method (Freuden-thaler, 2016) in agreement with EARLINET standards. For this purpose, a motorized filter wheel is implemented in the re-





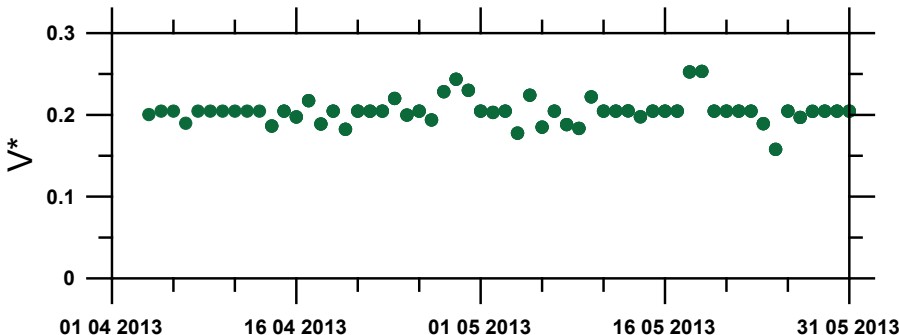

**Figure 2.** Daily depolarization calibration factor as derived during HOPE.

ceiver unit of Polly$^{\mathrm{XT}}$ to perform the $\Delta 90°$-calibration automatically three times a day. The procedure delivers the calibration constant $V^*$, which was found to be very stable for HOPE as shown in Fig. 2. It relies on the ratio of two signals and thus is invariant against most changes in the lidar setup (e.g., laser power, overlap, etc). For days with inappropriate weather conditions a standard value (mean of HOPE) is used. Only changes in the neutral-density filter setup of the polarization channels

will affect the depolarization calibration constant which was not the case during HOPE. Thus, we consider the calibration as very accurate with a standard deviation of less than 8% as seen in Fig. 2. By knowing the lidar-system-dependent transmission ratios $D_{\mathrm{c}}$ and $D_{\mathrm{tot}}$ (see Engelmann et al., 2016) of the cross and total channel, respectively, the volume linear depolarization ratio is derived without any further assumptions by

$$\delta^\lambda_{\mathrm{vol}}(R) = \frac{V^* - \delta^\lambda(R)}{\delta^\lambda(R)D_{\mathrm{tot}} - V^*D_{\mathrm{c}}} \tag{4}$$

with

$$\delta^\lambda(R) = \frac{P^\lambda_{\mathrm{c}}(R)}{P^\lambda_{\mathrm{tot}}(R)}, \tag{5}$$

where $P^\lambda_{\mathrm{c}}$ and $^\lambda P_{\mathrm{tot}}$ are the cross-polarized and total lidar signals, respectively. In the case of HOPE, depolarization measurements are available at $\lambda = 532$ nm.

### 3.3  Aerosol characterization

The methodology to derive the lidar system parameters was based on 30-min averaged profiles of the particle backscatter coefficient which are only available for specific atmospheric conditions. For the aerosol characterization aimed at in this paper, 24-hour measurements with 5-min resolution are used to characterize aerosols and clouds. The received signals of the backscattered light at 532 nm and 1064 nm and the depolarization ratio are used for this purpose. In the following, the methodology is introduced and then explained in detail in terms of a case study from HOPE.



### 3.3.1 Obtaining aerosol products - extensive properties

Since the molecular backscatter and extinction coefficients can be calculated from temperature and pressure profiles and the lidar system parameter can be estimated as described above, only the particle extinction coefficient, i.e. the transmission through the atmosphere, is left as an unknown in Eq. 1 to retrieve the particle backscatter coefficient. As a first guess for the particle
backscatter coefficient, the particulate attenuation in the atmosphere is neglected which reduces Eq. 3 to:

$$^{\mathrm{quasi^*}}\beta_{\mathrm{par}}^{\lambda}(R) = \beta_{\mathrm{att}}^{\lambda}(R)\exp\left\{2\int_{0}^{R}\alpha_{\mathrm{mol}}^{\lambda}(r)\,\mathrm{d}r\right\} - \beta_{\mathrm{mol}}^{\lambda}(R). \tag{6}$$

To account for the incomplete overlap of the lidar system in lower heights, an overlap correction function is applied and height-independent backscattering below 500 m is assumed in analogy to the calculation of the lidar system parameter $C^{\lambda}$. The particle extinction coefficient is now estimated in analogy to the procedure during the calculation of $C^{\lambda}$ by multiplying
$^{\mathrm{quasi^*}}\beta_{\mathrm{par}}^{\lambda}(R)$ with a constant lidar ratio of $S_{\mathrm{par}} = 55$ sr:

$$^{\mathrm{quasi}}\alpha_{\mathrm{par}}^{\lambda}(r) = {}^{\mathrm{quasi^*}}\beta_{\mathrm{par}}^{\lambda}(R)\,S_{\mathrm{par}}. \tag{7}$$

As explained already in Sec. 3.1, the lidar ratio value used served as a good compromise for lidar ratio values observed during HOPE and at other European continental sites. Finally, temporally high-resolved profiles of the so-called quasi particle backscatter coefficient defined as

$$^{\mathrm{quasi}}\beta_{\mathrm{par}}^{\lambda}(R) = \beta_{\mathrm{att}}^{\lambda}(R)\exp\left\{2\int_{0}^{R}\left[\alpha_{\mathrm{mol}}^{\lambda}(r) + {}^{\mathrm{quasi}}\alpha_{\mathrm{par}}^{\lambda}(r)\right]\mathrm{d}r\right\} - \beta_{\mathrm{mol}}^{\lambda}(R) \approx \beta_{\mathrm{par}}^{\lambda}(R) \tag{8}$$

can be calculated which serve as best estimate of the real particle backscatter coefficient $\beta_{\mathrm{par}}^{\lambda}(R)$ as demonstrated in Sec. 3.3.3. The quasi particle backscatter coefficient at 532 nm and 1064 nm is then used as the input for the particle characterization described below.

### 3.3.2 Obtaining aerosol products - intensive properties

With the calibration methods described above, a rough but temporally high resolved aerosol characterization can be done by using the quasi particle backscatter coefficients and the volume depolarization ratio to obtain intensive, i.e., aerosol-type specific quantities. From the quasi backscatter coefficients, the quasi Ångström exponent

$$^{\mathrm{quasi}}\mathring{a}_{\mathrm{par}}^{\lambda_1/\lambda_2} = -\frac{\ln\left(\frac{^{\mathrm{quasi}}\beta_{\mathrm{par}}^{\lambda_1}}{^{\mathrm{quasi}}\beta_{\mathrm{par}}^{\lambda_2}}\right)}{\ln\left(\frac{\lambda_1}{\lambda_2}\right)}, \tag{9}$$

is calculated for the wavelength pair $\lambda_1$ and $\lambda_2$, e.g. 532 nm and 1064 nm, to obtain information on particle size.
The quasi particle depolarization ratio defined as

$$^{\mathrm{quasi}}\delta_{\mathrm{par}}^{\lambda}(R) = \left[\delta_{\mathrm{vol}}^{\lambda}(R) + 1\right]\left(\frac{\beta_{\mathrm{mol}}^{\lambda}(R)\left[\delta_{\mathrm{mol}}^{\lambda} - \delta_{\mathrm{vol}}^{\lambda}(R)\right]}{^{\mathrm{quasi}}\beta_{\mathrm{par}}^{\lambda}(R)\left[1 + \delta_{\mathrm{mol}}^{\lambda}\right]} + 1\right)^{-1} - 1, \tag{10}$$





is also an intensive property and used to obtain information about the particle shape. The molecular depolarization ratio $\delta_{\mathrm{mol}}^{\lambda}$ is calculated theoretically from the bandwidth of the interference filters and is 0.0053 at 532 nm in case of Polly$^{\mathrm{XT}}$.

### 3.3.3  Example observation: 22 April 2013

To demonstrate the introduced quantities, the time-height cross sections of the four possible extensive (Fig. 3 shows) and four

possible intensive (Fig. 4 particle quantities of Polly$_{\mathrm{IfT}}^{\mathrm{XT}}$ are shown for one day of HOPE, the 22 April 2013.

The daily mean AOD was 0.34 at 500-nm wavelength and thus comparably high (monthly mean is 0.19). The atmospheric features are very well seen at 1064 nm and 532 nm while at 355 nm the atmospheric conditions are obviously not well represented which will be explained later in detail. The 22 April 2013started with a stratiform cloud with its base between 1.5 and 2.5 km which prevailed until 4 UTC. Below the cloud, inhomogeneous aerosol structures can be seen. The cloud is

characterized by a high quasi backscatter coefficient at all wavelengths and high volume depolarization ratio. After 4 UTC, a cloud-free nocturnal residual layer was observed. Note the layer structure which indicates a slow descent of the lofted aerosol layer. At around 10 UTC (12:00 lt), finally the growth of the convective PBL could be observed. The PBL reached up to 2–2.5 km on this day. At 20 UTC, the nocturnal PBL began to form as can be seen below 1 km height in Fig. 3. No low-level or mid-led level clouds were observed after 4 UTC, but cirrus at altitudes above 6 km (not shown) appeared after 13 UTC.

From the temporal development of the quasi particle depolarization ratio one can see a layer of enhanced depolarization mixed into the PBL from shortly after 12 UTC with a maximum at 1630 UTC. Obviously, non-spherical particles are mixed from the surface into the PBL and are dispersed as will be discussed further below. The particle depolarization ratio (Fig. 4) is also enhanced at the lower cloud boundaries due to multiple scattering and/or because of falling ice crystals.

The three Ångstöm exponents (Fig. 4) show a very different behaviour for which the Ångström exponents incorporating the

quasi backscatter coefficient at 355 nm are not representative. This is due to the corrections and assumptions made to estimate the particulate extinction and finally the quasi particle backscatter. As at 355 nm molecular backscattering is 80 (5) times higher than at 1064 (532) nm, large uncertainties are introduced into the attenuation correction presented in Sec 3.3.1 when 355-nm signals are considered, even though the lidar system parameter is known with good accuracy. The partial neglection of particulate extinction in the first-guess profile (Eq. 6) and the subtraction of the molecular scattering contribution leads often

to very large errors (as molecular backscattering is usually much stronger than particle backscattering at this wavelength) with even negative quasi particle backscatter coefficients. These effects are illustrated in Fig. 5 for a 30-min period of 22 April 2013. The real particle backscatter coefficients, the attenuated backscatter coefficient, and the quasi particle backscatter coefficients are shown for the different wavelengths.

We have considered also other approaches to estimate extinction for the calculation of the quasi particle backscatter coefficient

(cp. Eq. 8) at 355 nm. For example, by using the Ångström relationship (Ångström, 1964) to convert the 1064-m extinction with an assumed a-priori extinction-related Ångström exponent to the extinction coefficient profile at 355 nm similar to Eq. 9. Three different Ångtsröm exponents were chosen which are representative for the HOPE campaign, i.e. 1.0, 1.4, and 2.0, to obtain the extinction at lower wavelengths from the extinction at 1064 nm. This procedure is illustrated also in Fig. 5, where additionally the three backscatter coefficient profiles derived with this methodology are plotted. But also with that approach





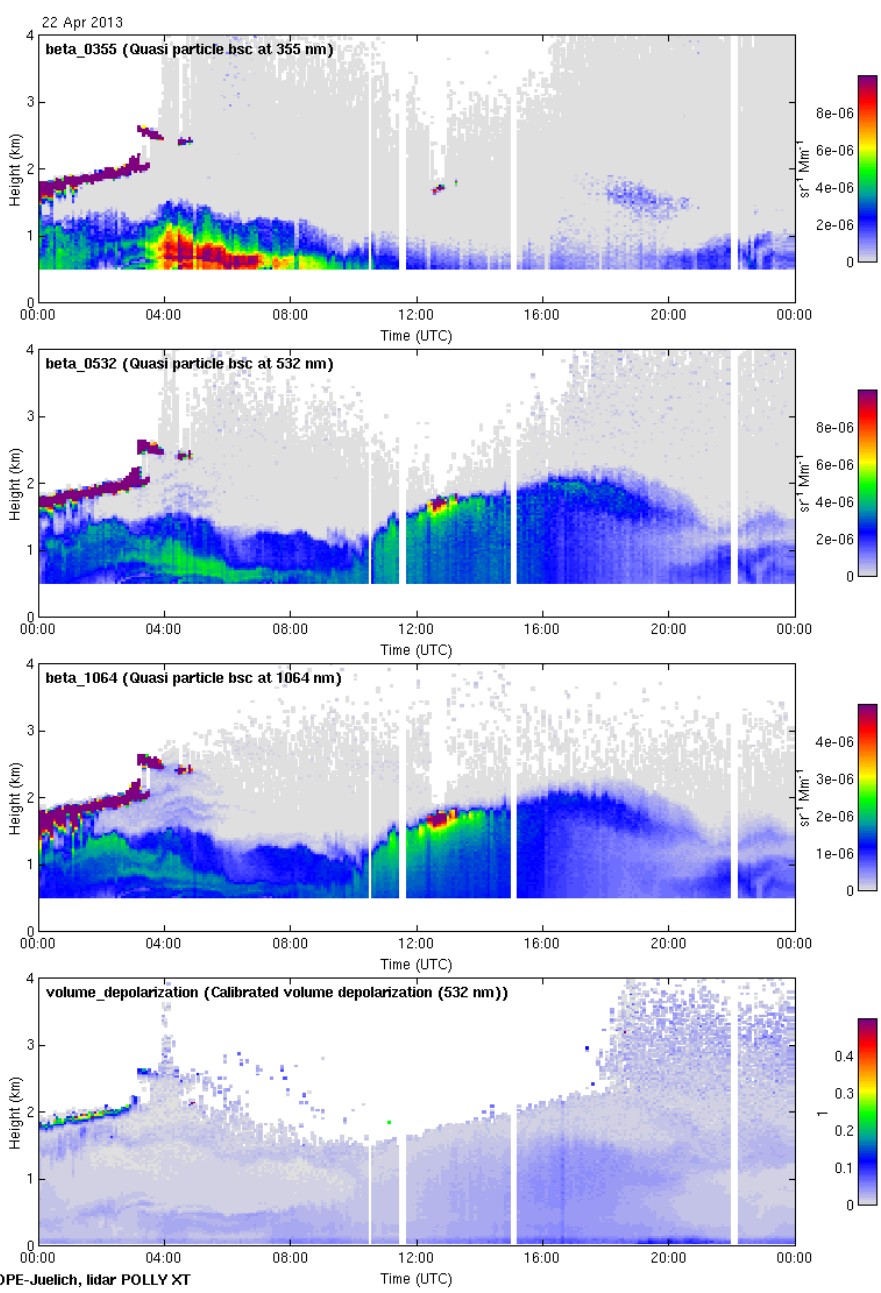

**Figure 3.** Polly observations at Krauthausen on 22 April 2013. Extensive properties from top to bottom: Quasi particle backscatter coefficient at 355 nm, 532 nm, 1064 nm, and volume depolarization ratio at 532 nm.



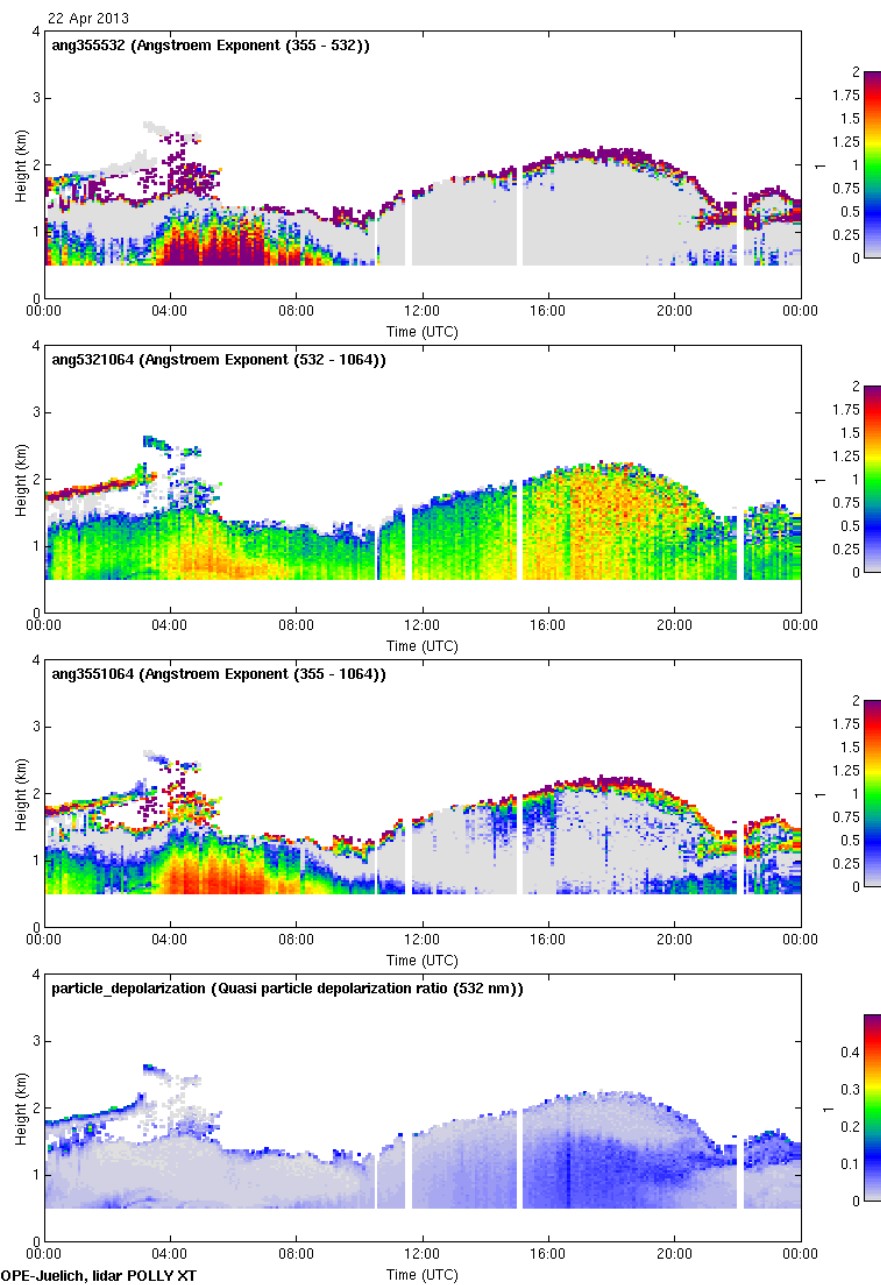

**Figure 4.** Polly observations at Krauthausen on 22 April 2013. Intensive properties from top to bottom: Quasi Ångström exponent for the wavelength pairs as indicated and quasi particle depolarization ratio.





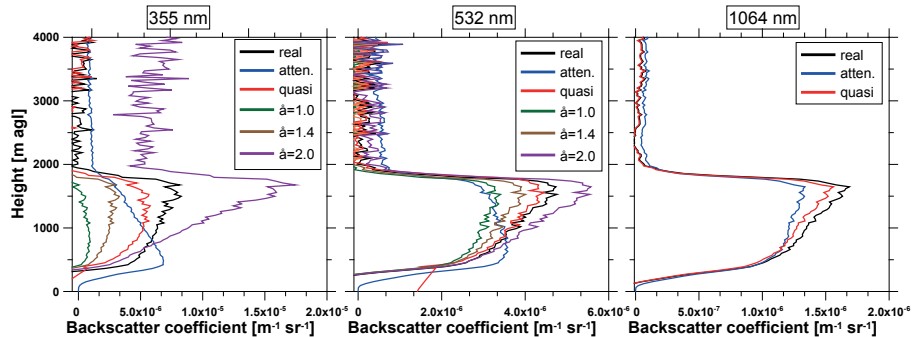

**Figure 5.** Comparison of the real particle backscatter, attenuated backscatter, quasi particle backscatter coefficient as defined for the three laser wavelengths for 22 April 2013, 1420–1450 UTC. Additionally, the quasi particle backscatter coefficient with a different approach for attenuation correction (Extinction coefficient derived from the 1064 nm extinction coefficient with the Ångström relation) is plotted for Ångström exponents of 1.0, 1.4, and 2.0

it was found, that the a priori choice of the extinction-related Ångström exponent is so crucial for 355 nm, that it cannot be applied in an automatic retrieval (e.g., see profile derived with an Ångström exponent of 2.0 at 355 nm). Closest to the real particle backscatter coefficient at all wavelengths is the quasi particle backscatter coefficient derived with the methodology described in Sec. 3.3.1 (without Ångström exponent assumption for extinction estimation). Taking into account the satisfying

results at 1064 nm and 532 nm with this approach, one can conclude that the quasi particle backscatter coefficient is a better estimate than the attenuated backscatter coefficient for particle backscattering in the atmosphere

This finding is also proved when comparing the different Ångström exponents as done in Fig. 6. Here, the Ångström exponent derived from the quasi backscatter coefficients at 532 and 1064 nm (yellow) is very similar to the truth (black, blue and red, all close to 1.4 and height independent for the aerosol layer up to 2 km) whereas the Ångström exponents using the 355-nm

quasi backscatter coefficient show already significant deviations (dark green and cyan). Even worse are the results when the attenuated backscatter coefficients are used (light green, orange, purple), which shows again that this quantity cannot be used for typing by using multiple wavelengths.

Consequently, we apply the quasi backscatter coefficients at 532 and 1064 nm, which are robust to determine and which are close to the atmospheric truth, the corresponding quasi Ångström exponent and the quasi particle depolarization ratio at 532 nm

for the temporally high-resolution target categorization.

## 4  Typing

For the typing of atmospheric features, i.e., the optical dominant scatterer type, three extensive (quasi backscatter coefficient at 532 and 1064 nm, and volume depolarization ratio) and two intensive properties (quasi Ångström exponent and quasi particle depolarization ratio) are available to detect aerosol and cloud layers and to distinguish between those two and classify





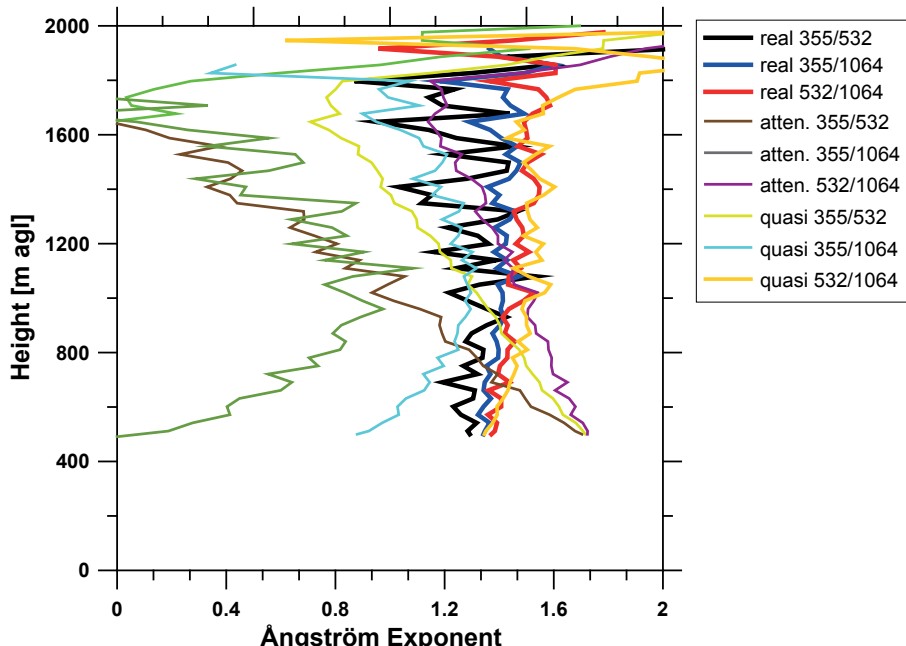

**Figure 6.** Comparison of Ångström exponents derived from the real particle backscatter, the attenuated backscatter, and the quasi particle backscatter coefficients for 22 April 2013, 1420–1450 UTC.

subtypes. A lidar-only attempt is made to categorize the aerosol and clouds concerning different types in analogy to the Cloudnet classification. In the following, the methodology is described followed by an intensive discussion concerning the applicability by means of example cases of HOPE.

### 4.1 Typing Methodology

5  The complete typing procedure based on the quasi backscatter coefficients, depolarization ratios, and Ångström exponent profiles is illustrated in Fig. 7 and listed in Tab. 1.

The lidar-only classification consists of the following main particle classes: non-typed particles, non-typed clouds, small spherical particles, large spherical particles, aerosol mixture, non-spherical particles, ice clouds, and liquid clouds. The "clean atmosphere" class represents a Rayleigh atmosphere where pure molecular scattering can be assumed. As the a priori information

10  used to derive the quasi backscatter coefficient (i.e. the lidar ratio assumed) are valid for aerosol particles only, we do not aim for making a complete cloud classification. However, the quantities available for typing are mostly representative to identify the bases of ice clouds and liquid clouds. Attenuation correction at the base is not crucial so the assumption of a wrong lidar ratio does not play a major role. However, we do not attempt to identify any particle classes above a liquid cloud as the attenuation correction will be corrupted.





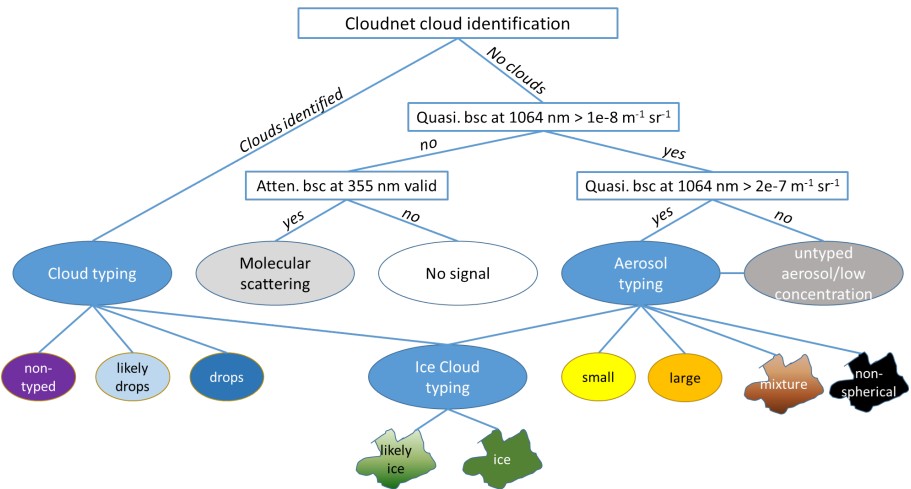

**Figure 7.** Schematical illustration of typing procedure. Details in text and Table 1.

**Table 1.** Overview of particle typing. Criteria for the feature classes are given. Quasi backscatter coefficient values are given in $\mathrm{m}^{-1}\,\mathrm{sr}^{-1}$.

| Detected feature | | | |
|---|---|---|---|
| Clean atmosphere | $^{\mathrm{quasi}}\beta_{\mathrm{par}}^{1064} \leq 1 \times 10^{-8}$ | | |
| Non-typed particles/low concentration | $^{\mathrm{quasi}}\beta_{\mathrm{par}}^{1064} > 1 \times 10^{-8}$ | | |
| Aerosol: small | $^{\mathrm{quasi}}\beta_{\mathrm{par}}^{1064} > 2 \times 10^{-7}$ | $^{\mathrm{quasi}}\delta_{\mathrm{par}} < 0.07$ | $\mathring{A}_{532-1064} \geq 0.75$ |
| Aerosol: large, spherical | $^{\mathrm{quasi}}\beta_{\mathrm{par}}^{1064} > 2 \times 10^{-7}$ | $^{\mathrm{quasi}}\delta_{\mathrm{par}} < 0.07$ | $\mathring{A}_{532-1064} < 0.75$ |
| Aerosol: mixture, partly non-spherical | $^{\mathrm{quasi}}\beta_{\mathrm{par}}^{1064} > 2 \times 10^{-7}$ | $0.07 \leq {}^{\mathrm{quasi}}\delta_{\mathrm{par}} < 0.20$ | |
| Aerosol: large, non-spherical | $^{\mathrm{quasi}}\beta_{\mathrm{par}}^{1064} > 2 \times 10^{-7}$ | $^{\mathrm{quasi}}\delta_{\mathrm{par}} \geq 0.20$ | |
| Cloud: non-typed | Cloudnet algorithm | | |
| Cloud: likely water droplets | Cloudnet algorithm | $^{\mathrm{quasi}}\delta_{\mathrm{par}} \leq 0.05$ | |
| Cloud: water droplets | Cloudnet algorithm | $^{\mathrm{quasi}}\delta_{\mathrm{par}} \leq 0.05$ | $\mathring{A}_{532-1064} \leq 0.5$ |
| Cloud: likely ice crystals | $^{\mathrm{quasi}}\beta_{\mathrm{par}}^{1064} > 2 \times 10^{-7}$ | $^{\mathrm{quasi}}\delta_{\mathrm{vol}} \geq 0.30$ | $^{\mathrm{quasi}}\beta_{\mathrm{par}}^{532} > 2 \times 10^{-7}$ |
| Cloud: ice crystals | $^{\mathrm{quasi}}\beta_{\mathrm{par}}^{1064} > 2 \times 10^{-7}$ | $^{\mathrm{quasi}}\delta_{\mathrm{par}} \geq 0.35$ | $^{\mathrm{quasi}}\beta_{\mathrm{par}}^{532} > 2 \times 10^{-7}$ |



Optical thick clouds are identified using the Cloudnet scheme for droplet finding (Illingworth et al., 2007; Hogan and O'Connor, 2004). As the lidar cannot penetrate liquid clouds, we cannot detect the cloud top in contrast to Cloudnet which uses the cloud radar information to gather this value. Therefore, the lidar target categorization will detect the cloud base and hydrometeors some tens of meters above the base. In principle within this scheme, clouds are detected if the backscatter coefficient at 1064 nm

is higher than $2e\text{-}5$ $m^{-1}$ $sr^{-1}$ and the signal decreases by a factor of 10 within 250 m above the maximum backscatter value. This algorithm is applied profile by profile and the corresponding pixels above the threshold are flagged as untyped cloud. If additionally the quasi particle depolarization ratio is below 0.05 they are flagged as most-likely droplets, while if also the Ångström exponent is less than 0.5 the pixels are flagged as droplets. The quasi backscatter coefficient threshold for clouds of $2e\text{-}5$ $m^{-1}$ $sr^{-1}$ account for an extinction coefficient of about $3.6e\text{-}4$ $m^{-1}$ at all wavelengths (Ångström exponent of 0 for big

particles e.g. water drops, lidar ratio of 18 sr). According to the OPAC data base (Hess et al., 1998), an extinction coefficient value of $3.6e\text{-}4$ $m^{-1}$ $sr^{-1}$ m is higher than the values at 550 nm given for all aerosol types except for strong pollution. According to Liu et al. (2009), a threshold of $1e\text{-}5$ $m^{-1}$ $sr^{-1}$ at 1064 nm is well suited for the discrimination between cloud and aerosol because the largest overlap between the two types is between $4e\text{-}6$ $m^{-1}$ $sr^{-1}$ and $1e\text{-}5$ $m^{-1}$ $sr^{-1}$. The automatically retrieved particle backscatter coefficient profiles as presented in Baars et al. (2016) showed that during HOPE aerosol particle backscat-

ter coefficients did not exceed $1e\text{-}5$ $m^{-1}$ $sr^{-1}$ (95% percentile maximum at $3e\text{-}6$). Thus, we consider the chosen threshold as valid for the conditions during HOPE without overlapping of the categories. Visual inspection showed no mis-classification of liquid clouds which convinces us that the approach is valid for the detection of cloud bases. As soon as liquid or untyped cloud is classified, no other classes above are evaluated because of the risk of strong attenuation, multiple scattering, etc., which disturb the signals significantly as the lidar applied is designed for aerosol and not for cloud detection.

We consider clean atmosphere if the quasi particle backscatter coefficient at 1064 nm is less than $1e\text{-}8$ $m^{-1}$ $sr^{-1}$ and a valid signal of the 355-nm quasi backscatter coefficient (SNR>0.5 at raw resolution of 30 s) is present. This threshold yields a ratio of molecular to particle backscattering at 532 (355) nm higher than 60 (180) at sea level and thus is valid for a Rayleigh calibration by means of the Raman or Klett-Fernald lidar method which might be one future application of the target categorization presented herein. The threshold of $1e\text{-}8$ $m^{-1}$ $sr^{-1}$ is also well below the given range for aerosols according to Winker et al.

(2009) for the CALIPSO classification. As the Polly$^{\mathrm{XT}}$ systems have a higher detection sensitivity than CALIPSO we cannot consider a higher threshold for clean atmosphere with Rayleigh scattering only. Anything above this threshold is first classified as untyped particle, which could be aerosol or clouds.

Aerosol and ice clouds are typed for a quasi backscatter coefficient at 1064 nm greater than $2e\text{-}7$ $m^{-1}$ $sr^{-1}$. Everything below remains classified as "untyped particle/low concentration". The threshold is equivalent to the one used in the CALIPSO feature

mask ($5e\text{-}7$ $m^{-1}$ $sr^{-1}$ for the 532-nm attenuated backscatter coefficient, Omar et al., 2009) when considering an Ångström exponent of 1.4 as measured as mean by AERONET during HOPE.

If the quasi particle depolarization ratio is less than 0.07 and the quasi Ångström exponent $\geq 0.75$, the scatterers are considered to be small particles. If the Ångström exponent is lower, it is supposed that large particles are dominating. A mixture of non-spherical and spherical particles is considered when the particle depolarization ratio is between 0.07 and 0.2, while above

0.2 the particles are categorized as large and non-spherical. The thresholds for the aerosols are chosen according to the work



of Amiridis et al. (2015) and Schwarz (2016), for which the analysis of observations of several EARLINET stations yields that large particles (marine, dust) have an Ångström exponent (532–1064 nm) less than 0.75 while smaller particle types (smoke, polluted continental, etc.) have an Ångström exponent (532–1064 nm) larger than 0.75. Pure Saharan dust is supposed to have a particle depolarization ratio at 532 nm of 31% (Tesche et al., 2009b; Ansmann et al., 2011) but also less was observed (around

28% e.g. Baars et al. 2016). Therefore, we consider particle depolarization ratios higher than 20% as mostly containing dust (or other non-spherical particles) and thus classify the scatterers as large, non-spherical particles. According to Tesche et al. (2009a), 20% particle depolarization ratio corresponds to a dust fraction in terms of backscattering of more than two thirds. A particle depolarization ratio of 7% on the other hand, corresponds to a dust fraction of less than 20%.

In contrast to other classification schemes (e.g., CALIPSO, Omar et al. 2009; HSRL, Burton et al. 2012), we do not categorize

by aerosol origin (e.g., mineral dust, biomass burning smoke, etc.) but by physical features. For example, large, non-spherical particles are in most cases mineral dust advected to the site but could be also volcanic ash, pollen or local dust plumes. The interpretation is not possible without additional information and thus will be left to the user of the categorization. We want to focus on the physical properties as these are the quantities we can obtain with this lidar-only approach.

Ice crystals, as they occur in cirrus clouds or virgae, are identified by their highly depolarizing properties independent of the

cloud identification or the aerosol typing and thus may overwrite these classes. As cirrus may be optically very thin, the same backscatter coefficient threshold as for aerosol is used to find ice crystals. The class "likely ice" is identified if the volume depolarization ratio (independent of quasi backscatter coefficient) is higher than 30%. "Ice crystals" are identified if the particle depolarization ratio is higher than 35% and may overwrite the "likely ice" class. However, the identification of ice crystals is the most critical matter, as sometimes the depolarization information at 532 nm is not available due to the low SNR whereas with

1064 nm channel these particles can be detected. Thus, many ice crystals remain unclassified and are categorized as untyped particles or clouds.

In the next section, we want to demonstrate the performance of the newly developed target categorization by means of three example cases.

### 4.2 Examples for the aerosol categorization

In the following, the observation days of 22nd, 4th, and 18th of April during HOPE are discussed by means of the lidar target categorization.

#### 4.2.1 22 April 2013

Figure 8 shows the newly developed classification scheme for MWL for the example day of 22 April 2013presented in Section 3.3.3. Several features were successfully detected: Between 00 and 04 UTC, the cloud base of the liquid cloud was

successfully identified. The base was categorized as "Cloud: likely water droplets light" (blue). Due to the required a priori assumptions for the quasi backscatter coefficients which are aiming at aerosols, the Ångström exponent was not below 0.5 and thus the "Cloud: water droplets" requirements were not fulfilled. Above the cloud base, the depolarization ratio is slightly enhanced due to multiple scattering (see Fig. 3, bottom) and thus the cloud is classified as "untyped cloud". Below the cloud, at



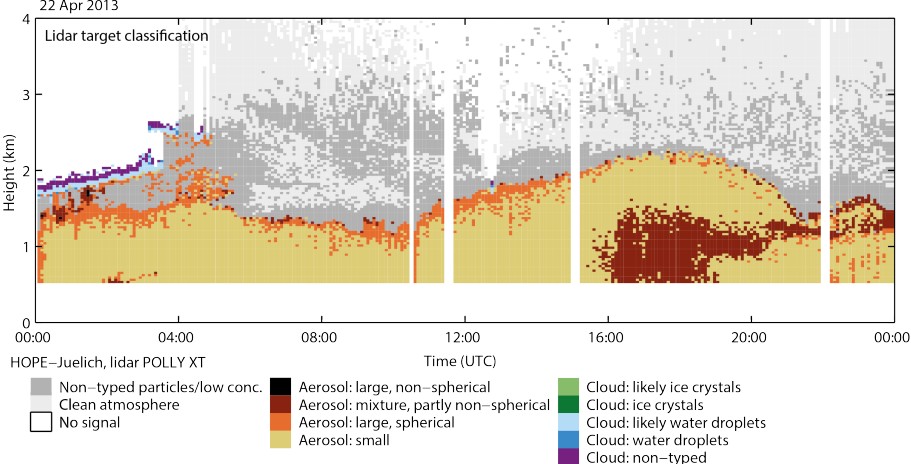

**Figure 8.** Lidar particle categorization for 22 April 2013.

the top of the PBL, large aerosol (orange) is identified above small aerosol particles (yellow) due to the low Ångstrom exponent (532/1064nm, see Fig. 4). The growth of aerosol with increasing altitude within the PBL is most probably caused by hygroscopic growth. After 4 UTC, the cloud deck dissolved and an aerosol layer with mostly small particles but large particles at the top remains the whole day. A small cumulus cloud was observed in addition shortly past 12 UTC at the top of the convective

PBL remaining the only cloud at daytime on this day. The aerosol layer top and thus also the PBL top reached its maximum with 2.2 km at around 19 UTC before the aerosol layer starts to decay. We have to note that from the lidar target categorization the identification of the PBL, i.e. the mixing layer height, is not possible and needs additional information, therefore we refer with the term PBL to the main aerosol layer which might have been very often coincided with the mixing layer during daytime. An interesting feature is the entrainment of partly non-spherical particles (brown) between 16 and 19 UTC from the surface.

After 19 UTC, these non-spherical particles were detected close to the top of the nocturnal aerosol layer. The source of these non-spherical particles could be local dust (from open-pit mining close by, see Macke et al. 2016 for a map) and/or pollen from the local agricultural plants (e.g. see Fig.1b in Maurer et al. 2016). Such entrainment from ground was very often observed in April at Krauthausen and needs to be investigated further in future. Above the main aerosol layer, some aerosol but in low concentration is identified (dark grey) which means that this regions are not suitable for the so-called Rayleigh fit (Freuden-

thaler, 2009) needed for the Raman or Klett-Fernald lidar method for which one needs regions of molecular scattering only (light grey).

For comparison, Figure 9 shows the standard Cloudnet classification (Illingworth et al., 2007) which is derived from cloud radar, microwave radiometer, and ceilometer observations. This classification allows us to distinguish between the different cloud types and to detect aerosol. However, no discrimination between aerosol types is possible. At around 2 km between 0

and 3 UTC, clearly a supercooled liquid layer was observed (slightly above the lidar detected cloud base). Below, ice crystals were identified which turned into liquid at about 1.2 km. According to temperature profiles retrieved from GDAS1 (Global





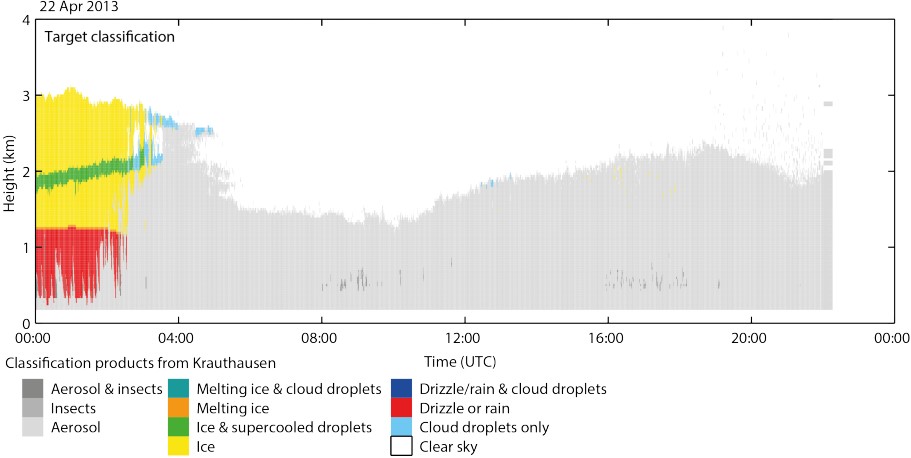

**Figure 9.** Cloudnet target categorization for 22 April 2013.

Data Assimilation System, https://www.ready.noaa.gov/gdas1.php) for the lidar location, the 0 °C altitude was 1.4 km confirming the findings. The identification of the liquid droplet layer by Cloudnet shows that the detected cloud features by lidar are certainly mostly liquid droplets and thus confirm the correct classification by the lidar categorization. The lidar however, did not identify drops or ice below the cloud most probably due to the low concentration of these hydrometeors for which the

lidar is not sensitive. After 4 UTC, Cloudnet classifies aerosol only. The small cloud layer as observed with the MWL is also detected shortly past 12 UTC.

Finally, we can conclude the lidar-only target categorization works well and is in agreement with Cloudnet even though the different instrumentations allow the detection of different atmospheric features.

### 4.2.2   4 April 2013

A second example case to be discussed is 4 April 2013at Krauthausen. The MWL target classification (top) and the Cloudnet ones (center and bottom: LACROS and JOYCE) are shown in Fig. 10. JOYCE (Jülich ObservatorY for Cloud Evolution, 3 km away) data is shown because no data from Cloudnet is available for LACROS past 17 UTC due to maintenance work on the cloud radar. Nevertheless, the most interesting feature on this day is the overcast cloud condition between 3 and 10 UTC. During this time, the MWL classification detects very well the cloud base (cloud or likely cloud) and large aerosol below. The

Cloudnet classification however, detects the liquid cloud base as well, but classifies below ice and super cooled droplets and/or ice not touching the ground. According to the temperature profile derived from the GDAS1 data set, a strong inversion was present between 1.8 and 2.2 km and temperatures were below 0° Celsius throughout the troposphere. Thus, both classifications are reasonable and one could suppose that the ice and drizzle detected by the radar led to evaporation which increased the relative humidity (RH) in the aerosol layer and led to hygroscopic growth and finally, as detected, to large, spherical aerosol

particles. As at the cloud base 100% RH can be considered, the particles just below the cloud experienced high RH and thus a





**Figure 10.** Lidar particle categorization (a) and Cloudnet target categorization (b and c) for 4 April 2013 at Krauthausen (a and b) and Jülich (c).





strong particle growth has most likely led to increased scattering (e.g., Skupin et al., 2014).

This example shows very illustrative the different sensitivity concerning particle size and thus the potential synergy between the lidar- and radar-based classification. While the lidar is more sensitive to the numerous but comparably small aerosol particles, the radar is most sensitive to the few but large precipitation particles. Therefore, we conclude that between 3 and 10 UTC

all detected features, i.e., large, spherical aerosol particles and ice and supercooled drops were present simultaneously. It also shows that under conditions of low-level clouds, atmospheric features can be identified by MWL with the newly developed methodology which is not easily possible with the traditional Raman or Klett-Fernald lidar methods.

Past 11 UTC, another cloud with its base at around 1 km was detected at the top of the growing PBL. Again, the cloud base is identified with lidar at the height at which Cloudnet identifies cloud droplets only. Above and below the cloud base, Cloudnet

classifies ice crystals which cannot be verified with the MWL target categorization. There, mostly small but also some large, spherical particles close to the cloud base are identified. Above the cloud base, no valid lidar signal is available.

Past 16 UTC, the lidar detected ice clouds from 2.5 to 6 km height which was observed with Cloudnet instrumentation, too. Cloudnet is able to detect the ice clouds already before at altitudes up to 9 km which is not possible with the MWL during the low-level-cloud-deck period. Interestingly during the period past 16 UTC, a lofted aerosol layer was found below the ice cloud

between 2 and 3 km classified mostly with spherical particles. Below, in the transition zone to the PBL, non-spherical particles were identified because of an increased depolarization ratio while in the PBL itself, small, spherical particles were observed. The Cloudnet observations from JOYCE only 3 km away, however, gave no indication of ice crystals at this altitude so that we can conclude that the non-spherical particles were advected towards the site.

Interestingly, at around 17 UTC, large, spherical particles are directly classified below/within the ice cloud at around 3.5 km

because of low depolarization values. We can only speculate that due to evaporation of ice crystals, residual aerosol might have grown. Unfortunately, the radar at LACROS was not in operation to investigate this feature in more detail.

### 4.2.3   18 April 2013

The third example day, 18 April 2013, is shown in Fig. 11. This day is characterized by strong westerly winds with wind-gusts up to 16 m/s as it was found from Doppler lidar observations. On this day, a mixture with non-spherical aerosol in the

lowermost boundary layer was observed almost continuously, except for the period of cloud occurrence between 5 and 8 UTC. This liquid cloud is identified with MWL and Cloudnet in good agreement. The MWL classification detects an optically thin lofted aerosol layer between 2 and 3.5 km height after the low cloud layer disappeared at around 7 UTC. Cloudnet did not detect this aerosol layer. At the top of this layer a cloud formed shortly past 8 UTC. Both clouds are identified to be pure liquid by both algorithms. Shallow boundary layer clouds were observed occasionally past 12 UTC.

Due to the strong westerly winds, we conclude that the observed non-spherical particles in the PBL origin from the open-pit mine of Inden (see, e.g., Fig. 2 in Macke et al. 2016) west of our measurement location. Usually, most of these particles remain below 1 km at the lidar site (except during the growing phase of the PBL from 10 to 12 UTC). This is an indication that the particles were just entrained into the PBL and did not had the time yet to be transported to the top of the PBL. Another reason could be that the particles were of much larger size than typical aeolian dust and thus sediment much more rapidly after their





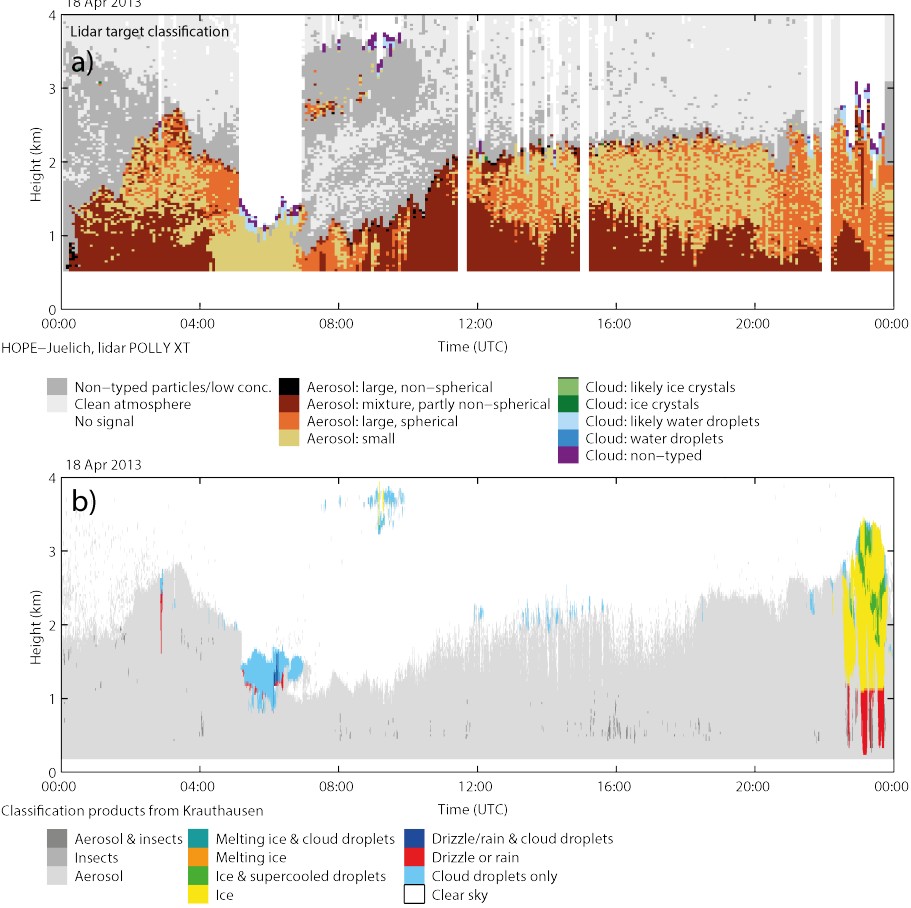

**Figure 11.** Lidar particle categorization (a) and Cloudnet target categorization (b) for 18 April 2013.

emission than other particle types. Visual inspection of the pit mine of Inden 1.5 km west of the LACROS site, proved strong dust emissions as shown in Fig. 12.

After 23 UTC, a shallow convective cloud system was observed whose precipitation (first ice than drizzle) did not touch the ground at the LACROS site (see Cloudnet categorization in Fig. 11). The MWL target categorization also detects the cloud

5  but as already discussed in the previous example case, does not resolve the drizzle and ice but identifies large aerosol particles which might again have been influenced by hygroscopic growth due to precipitation evaporation.

## 5  HOPE

In this section, an overview about the aerosol conditions during entire HOPE is provided. The MWL Polly$_{\text{IfT}}^{\text{XT}}$ was routinely operating at Krauthausen from 2 April 2013to 31 May 2013. Thus, two full months of a spring season could be covered. An

10  overview of the observations of the full campaign is given in the Appendix in Fig. 15 (April) and Fig. 16 (May) in terms of the




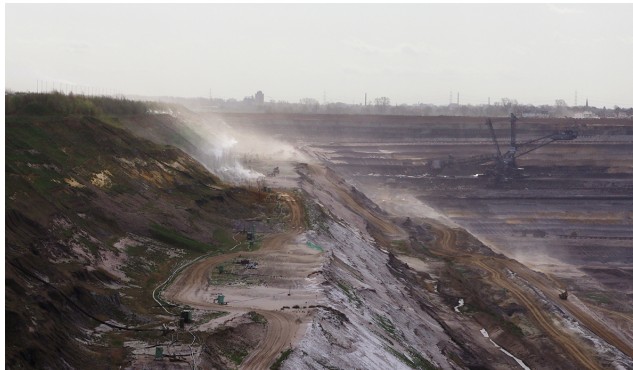

**Figure 12.** Photograph of the easterly border of the open-pit mine of Inden on 18 April 2013. Strong dust emissions were observed. The LACROS site was located 1.5 km east (i.e., downwind) of the pit.

quasi particle backscatter coefficients at 532 and 1064 nm (extensive properties), the quasi Ångström exponent (532–1064 nm), and the quasi particle depolarization ratio (intensive properties) as used for the categorization. As described in Macke et al. (2016), the weather conditions during HOPE varied from periods with several warm and cold front passages interrupted by a few high pressure systems with high level cirrus clouds at the beginning of the campaign to more low-level convective clouds

conditions later on.

Continuous MWL observations were available during the entire period with the exceptions of some short interruptions due to maintenance. During days of almost only precipitation (e.g. 16 May 2013), lidar observations are only sporadically available as the system stops measurements during precipitation events. Thus, calibrated lidar signals and the corresponding Ångström exponents and depolarization ratios are available for most of the time of favourable weather conditions and allow the typing of

the particles according to the scheme described above.

The corresponding lidar target categorization for entire HOPE aiming on aerosol discrimination is shown in Fig. 13 together with the respective Cloudnet classification. The lidar target categorization reveals that aerosol was usually located from the ground up to 2 km height. Non-typed particles and low aerosol concentration were typically detected up to higher altitudes (4-5 km) showing that these regions are not appropriate for the Rayleigh fit procedure as already described above. Further-

more, it can be seen that the spring 2013 at Krauthausen was dominated by low-level clouds and cirrus. Only on a few days, clear sky conditions were observed. Comparing to the Cloudnet target categorization, it is confirmed that April and May was often dominated by deep clouds covering almost the whole troposphere. The lidar target categorization does by definition only identify the cloud bottoms in these cases but this in good agreement with Cloudnet.

Interestingly, the intrusion of non-spherical particles was observed several times in the lowest 2 km until beginning of May (see

lidar target categorization in Fig. 13). We can only speculate that this might be local dust form open-pit mining, as intensively discussed for the 18 April case study, or pollen. After 10 May 2013, low-level clouds together with precipitation prevailed (see also Cloudnet target categorization) and thus it is reasonable that the local dust was too wet to be entrained into the air and/or the pollen season was over. These observations might be an interesting topic for future studies focussing on local aerosol





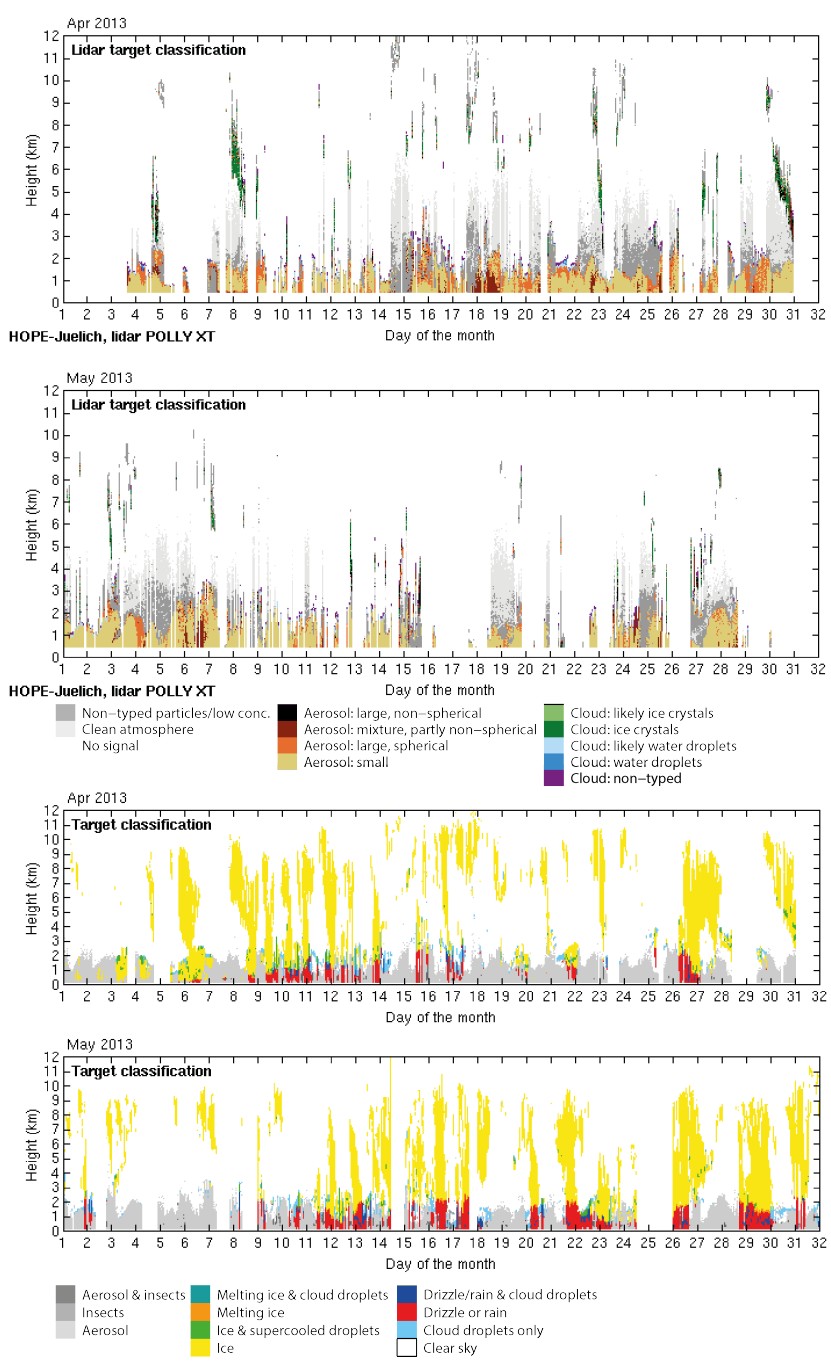

**Figure 13.** Lidar particle categorization (top) and Cloudnet categorization (bottom) for April and May 2013.




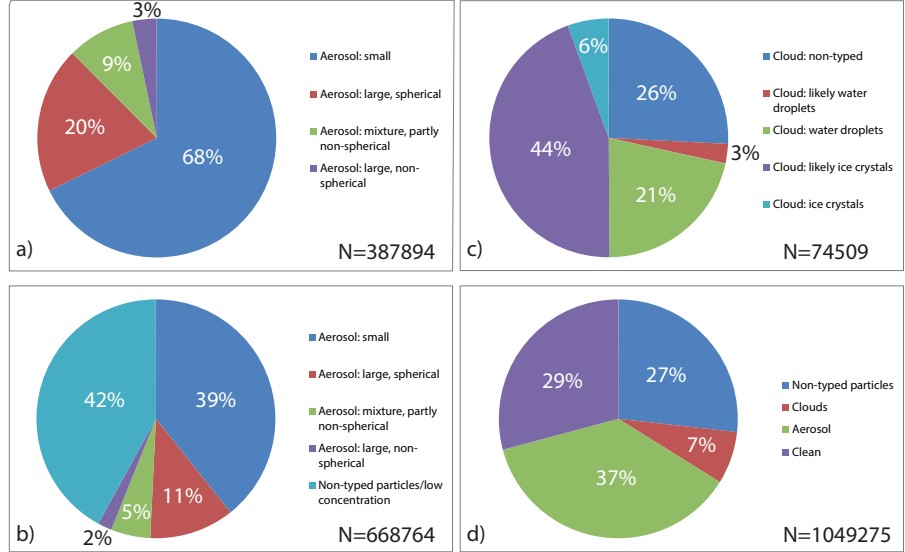

**Figure 14.** Statistics on particle categorization for the entire HOPE campaign: a) for all typed aerosol particles, b) typed aerosol and untyped particles, c) cloud particles, d) all typed pixels.

emissions.

Furthermore, one sees that during HOPE the majority of the aerosol in the PBL was classified as small aerosol as we would expect for an industrial and highly populated area. However, large aerosol was also observed occasionally, but mostly at the top of the PBL indicating hygroscopic growth. Comparing again to Cloudnet, one sees that often drizzle is observed with radar

while the lidar still detects aerosol. This interesting feature, discussed already for the presented case studies, was observed frequently and demonstrates the different sensitivity of the different instruments. Furthermore, it is found that Cloudnet does not detect as much aerosol with low concentrations due to the use of the ceilometer which is not as powerful as the MWL.

To give an overview of the aerosol and also partly the cloud conditions during HOPE, a statistics of the classified scatterers for the entire troposphere for HOPE is shown in Fig.14. Concerning typed aerosol (Fig.14, top, left), the majority of the particles

were classified as small aerosol (two thirds). Large, spherical particles were observed in 20% of the time, while a mixture of non-spherical and spherical particles was observed in 9% and large, non-spherical particles only in 3% of the analyzed pixels. As already discussed, mostly these particles were mixed from the ground into the atmosphere and only on a few days advected thin lofted layers with obviously Saharan dust were observed.

If one takes into account also the "non-typed particles/low concentration" class (Fig. 14, left, bottom) one sees that surpris-

15 ingly 42% of the particles are untyped or of low concentration. But one has to take into account, that this particle class can inherit every scatterer type, clouds, aerosol, etc. and that very low aerosol concentrations were almost always present above the PBL. Due to the conservative approach chosen, particles are only typed if enough information are available. Thus, often the 1064-nm-channel, which is least-sensitive for molecular scattering, detects particles but the other channels have a too low





SNR to be used for typing the particles which lead to the large number of untyped aerosol. However, the information given is still very useful, as it makes clear that not only molecules contribute to the light scattering, which is important when the target classification will once be used for the determination of suitable calibration periods and regions with negligible aerosol scattering.

For the clouds identified during HOPE, a different picture was obtained (Fig.14, top, right). Here, the "likely ice cloud" class is the dominant type with 46%. Due to the assumption made above aiming on aerosols (lidar ratio), the quasi particle depolarization ratio is underestimated in cirrus and thus does often not exceed 35%. Therefore, the clearly identified ice clouds make only a fraction of 6%. However, we have to repeat that we do not aim at classifying clouds as we focus on aerosol and the cloud information might be used only as a hint for the type of clouds for which further investigations are necessary. Water droplets

are typed in 21% of all cases and likely liquid clouds only in three percent of the time. Non-typed clouds amount for 26% of all cloud classes. We have to repeat that this cloud statistic is biased as the lidar can penetrate liquid clouds only by a few tens of meters. Above a detected liquid cloud no typing is performed. In turn, the lidar can often penetrate cirrus clouds and thus in contrast to liquid clouds, ice crystals can be detected also well above the cloud base.

All together during the HOPE campaign, more than 1 million pixels of 30 m vertical and 5 min temporal resolution were suc-

cessfully analysed. From these pixels, clean (i.e. molecular scattering dominating) atmosphere was observed in 29%, clouds in only 7%, aerosol in about 37% and non-typed particles/particles with low concentration in 27% of the analysed and feature-classified pixels (Fig.14, bottom, left).

## 6   Conclusions

In this work, we have used absolutely calibrated lidar signals to categorize primary aerosol but also clouds in high temporal

and spatial resolution. Two months of 24h/7days observations from the multiwavelengths-Raman-polarization lidar $\text{Polly}^{\text{XT}}_{\text{IfT}}$ during the HOPE campaign have been used for that purpose. We have used the well established Cloudnet framework to develop a lidar stand-alone classification. The Cloudnet equipment was operated continuously directly next to the lidar and have been used for comparison.

Automatically derived particle backscatter coefficient profiles (Baars et al., 2016) in low temporal resolution (30 min) have

been used to calibrate the lidar signals. A daily mean lidar calibration parameter was derived with an accuracy better than 20%. From this calibrated lidar signals, new atmospheric parameters in temporally high resolution (quasi particle backscatter coefficient) have been developed which require a priori information (assumptions) for attenuation correction. It was found that the newly developed procedure works well at 532 nm and 1064 nm but deviations from the real particle backscatter coefficients can be strong at 355 nm when the a priori information are not perfect. As a consequence for the particle typing, the quasi particle

coefficients at 532 nm and 1064 nm, its corresponding Ångström exponent, and the linear depolarization ratio at 532 nm are used for the classification.

By using thresholds obtained from multi-year, multi-site EARLINET measurements, four aerosol classes (small; large, spherical; large, non-spherical; mixed, partly non-spherical) are defined. Thus, particles were classified by their physical feature



(shape and size) instead of classifying them by source as, e.g., the well known CALIPSO typing does. For source definition additional information are needed which have been out of the scope of this development which have focussed on a lidar stand-alone tool.

The bases of optical thick clouds (liquid droplets) can be successfully identified using the Cloudnet approach. Cirrus clouds/ice are identified by its highly depolarizing features. Furthermore, regions dominated of molecular scattering and regions of untyped particles/low aerosol concentration are identified with the target categorization. The detection of molecular regions can be very useful for, e.g., lidar calibration in the atmosphere.

By discussing three 24-h case studies, it was shown that the aerosol discrimination is very feasible and informative and gives a good complement to the Cloudnet target categorization. By analysing the entire HOPE campaign, almost 1 million pixel (5 min, 30 m) could be successfully classified with the newly developed tool from the two months data set. We found that the majority of the aerosol, trapped in the PBL, were small particles as expected for a heavily populated and industrialized area. Large, spherical aerosol was found mostly at the top of the PBL and close to cloud bases indicating the importance of hygroscopic growth of the particles at high relative humidity. Interestingly, it was found that on several days non-spherical particles were intruded from the ground into the atmosphere. The origin of these particles remains unclear and needs further research. Lofted layers of Saharan dust as typical for spring in Germany were observed only sporadically and with low AOD during the investigated time frame of the HOPE campaign in spring 2013. Untyped aerosol with low concentrations was found often above the PBL up to heights of about 4 km. Cloudnet couldn't identify these optically thin particle layers due to the lower sensitivity of the used ceilometer. The capability to detect cloud bases was compared to the Cloudnet feature mask and the good agreement gives evidence that this feature could be used to apply robust cloud screening as often needed for lidar data retrievals., e.g., for other automatic approaches as the EARLINET Single Calculus Chain (D'Amico et al., 2015). Ice crystals were also often classified correctly, but sometimes remained unclassified or even false classified as aerosol as a consequence of multiple reasons (a priori information aiming at aerosol, low depolarizing characteristics in certain temperature ranges, etc.). This behaviour might be overcome when combining the lidar stand alone target categorization with the Cloudnet target categorization as planned in ACTRIS-2 (ACTRIS is the European Research Infrastructure for the observation of Aerosol, Clouds, and Trace gases. http://www.actris.eu). Then, the 10 lidar-based target types are available in addition to the already existing Cloudnet quantities for an advanced categorization of both aerosol and clouds. In this way, errors, i.e. mis-classifications, could be minimized in both schemes and a detailed data set could be provided for European and other supersites hosting both Cloudnet standard equipment and reliable, automatic, high-quality lidars based on EARLINET standards.

However, it is important to have a lidar stand alone tool, as at the moment only at three European stations Cloudnet and automatic continuously running MWL are operated, while stand-alone lidar systems are available at more than 25 EARLINET stations. We also consider the presented MWL approach for the classification of aerosol types as a prerequisite for the development of schemes for the identification of aerosol layers. Current retrievals, such as the STRAT algorithm (Morille et al., 2007), aim for providing aerosol layering information from lidar observations at one wavelength and can thus only identify a single layer even though it would actually consist of several layers of different types, such as smoke or dust. With this development,




the integration of EARLINET and Cloudnet is ongoing and offers a high potential for future synergistic profiling of aerosols, clouds and their interaction by combining modern state-of-the art atmospheric instruments.

*Acknowledgements.* The authors acknowledge support through the research program "High Definition Clouds and Precipitation for Climate Prediction – HD(CP)2" (FKZ: 01LK1209C and 01LK1212C) funded by Federal Ministry of Education and Research in Germany (BMBF),

5   ACTRIS under grant agreement no. 262254 and ITaRS under grant agreement no. 289923 of the European Union Seventh Framework Programme (FP7/2007-2013), and ACTRIS-2 under grant agreement no. 654109 from the European Union's Horizon 2020 research and innovation programme. Many improvements, both, in terms of hard and software were triggered by the fruitful discussions and network activities within EARLINET. The software framework of Cloudnet was used for this development for which the authors are grateful. We also acknowledge the use of JOYCE data which are provided via the HD(CP)² data portal.




## Appendix A: Measurement overview

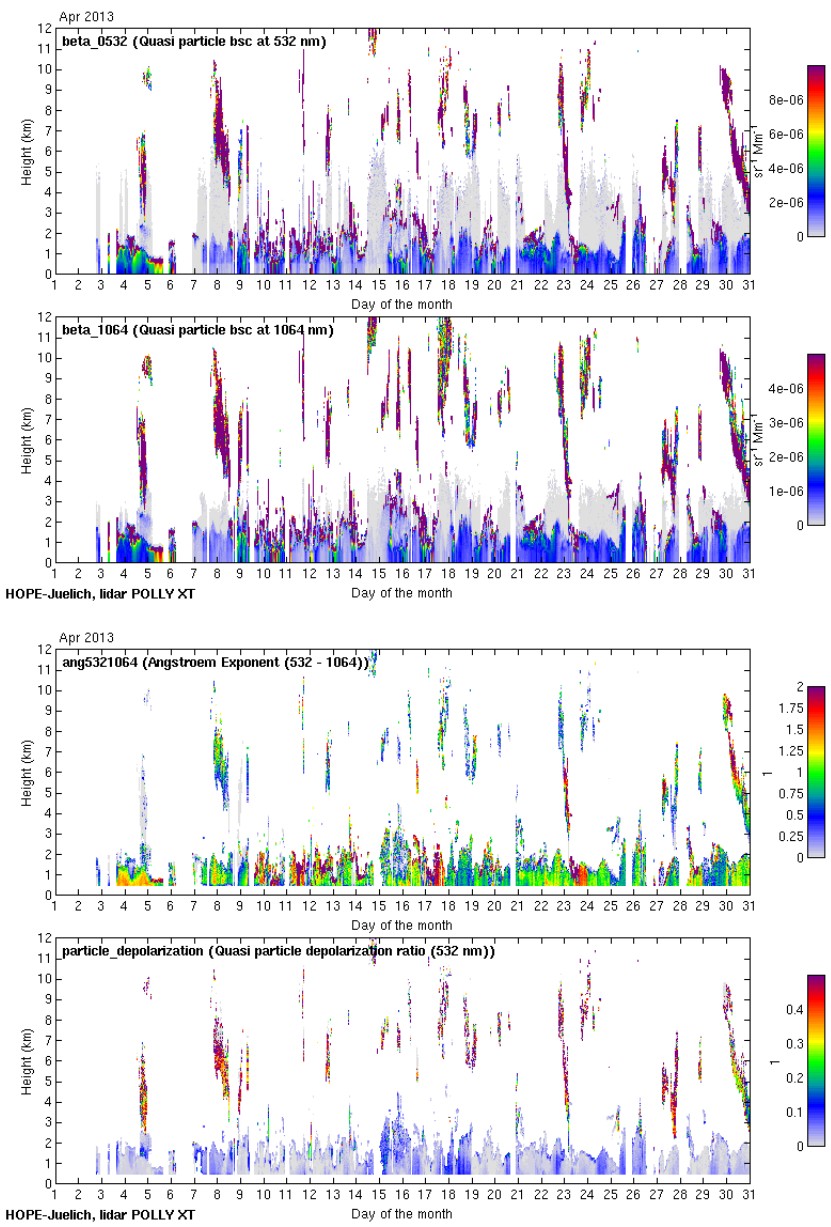

**Figure 15.** Overview of MWL Polly$^{\mathrm{XT}}$ observations in April 2013. Top to bottom: Quasi particle backscatter coefficient at 532 nm and 1064 nm, respective Ångström exponent, and quasi particle depolarization ratio at 532 nm.





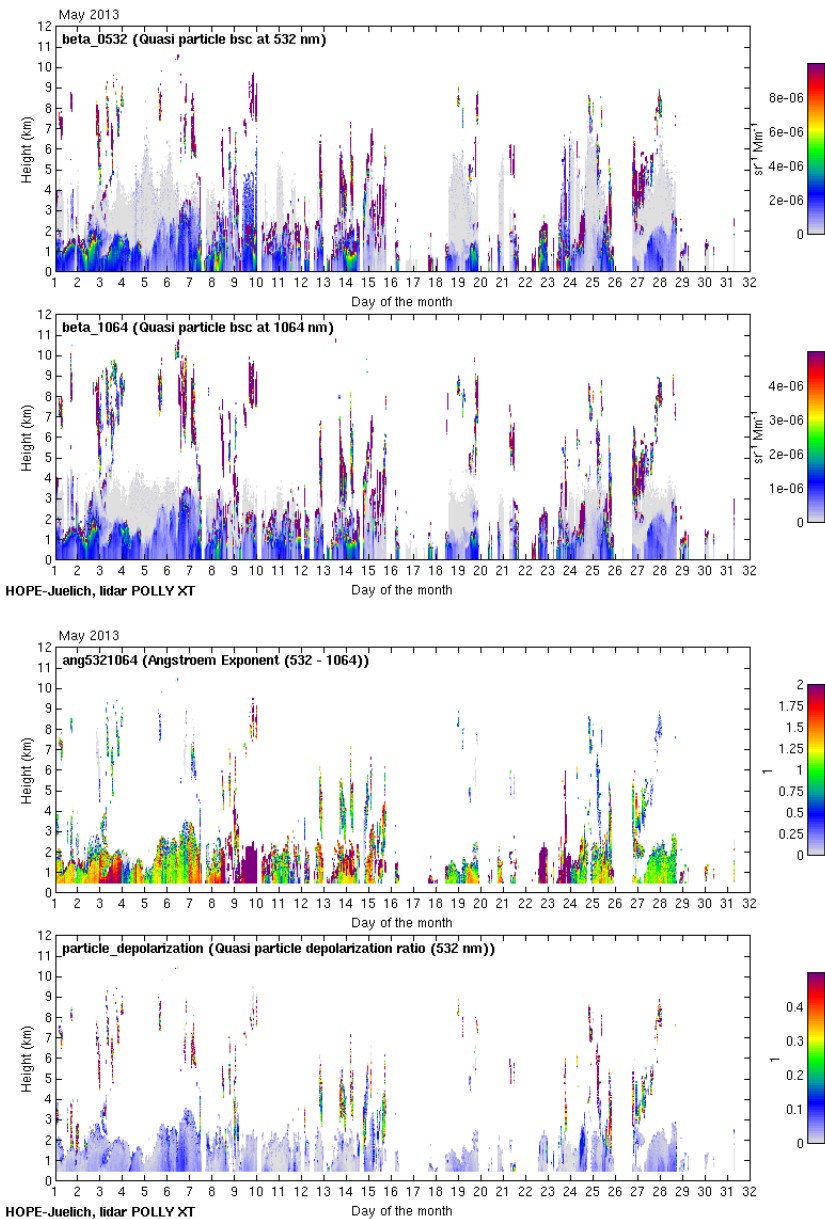

**Figure 16.** Overview of MWL Polly$^{\text{XT}}$ observations in May 2013. Top to bottom: Quasi particle backscatter coefficient at 532 nm and 1064 nm, respective Ångström exponent, and quasi particle depolarization ratio at 532 nm.



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
