# Peer review of "Target categorization of aerosol and clouds by continuous multiwavelength-polarization lidar measurements"

_Atmospheric Measurement Techniques, 2016_

## Referee Comment (RC1) · Anonymous Referee #1 · 21 Mar 2017

This is an interesting, well written paper proposing a methodology for categorizing on a physical basis aerosols and clouds (the latter within certain conditions and limits) with high temporal resolution from lidar-only data.

The methodology is illustrated with practical examples and appears to be adaptable to aerosol lidars with the capabilities of that operated by the authors, provided the instrument constant of the different channels exhibits a minimum stability. The mutiwavelength, polarization-sensitive type of lidar required to apply the methodology is certainly sophisticated, but such instruments are becoming increasingly common thanks to the expertise developed and disseminated through initiatives and networks such as EARLINET, PollyNET, GALION, etc.

Lidar-only categorizations are compared to Cloudnet ones, the authors pointing out the synergy obtained by the lidar- and radar-based methodologies.

The paper should be published, although its reading prompts some questions / remarks detailed in the following that the authors are requested to address.

**Main questions:**

1. One of the parameters used in the proposed categorization method is the so-called quasi-particle backscatter coefficients. Profiles of this parameter are obtained assuming a typical lidar ratio and correcting the attenuated backscatter for the molecular extinction and the particle extinction, the latter estimated by multiplying the lidar ratio by a "first guess" of the quasi-particle backscatter coefficient obtained from the attenuated backscatter corrected only for the molecular extinction. Eqs. (6)-(8) describe clearly the procedure. However one wonders if the cycle could not go on –  i.e. if the quasi-particle backscatter coefficient so obtained could not be used to refine the estimate of the particle extinction (Eq. (7)), which in turn would be inserted again in Eq. (8) to obtain a new estimate of the quasi-particle backscatter – until a convergence criterion is attained. It doesn't seem that implementing this iterative cycle would be too cumbersome from the computational point of view and it would probably converge to what the authors call the "real backscatter coefficient" on lines 293, 305, and 616, and in the captions of figs. 4 and 5. Have the authors tested if this would improve the performance of the categorization?

2. I find the term "real backscatter coefficient", as used where indicated in the above remark, somewhat misleading, because, even though the authors do not indicate explicitly how it is obtained (Klett-Fernal algorithm, iterative algorithm?), it relies probably on an assumed lidar ratio. The authors should clarify this.

3. In connection with the previous remark, the authors should also clarify what is understood by the "truth" on line 313.

4. On lines 14-16, it is stated: "By analyzing the entire HOPE campaign, almost 1 million pixel (5 min times 30 m) could be successfully classified from the two months data set with the newly developed tool". I wonder if the claim (repeated in a somewhat different and possibly less strong and more appropriate way on line 598-599: "more than 1 million pixels of 30 m vertical and 5 min temporal resolution were successfully analysed") is not too exaggerated. If we understand by "successfully classified" that the pixels actually contained the particle class assigned by the algorithm, can the authors be sure of that?

5. In the criteria for the categorization, clearly summarized in table 1, the Ångström exponents are not always taken into account (e.g. for the aerosol mixture, partly non-spherical, for large, non-spherical aerosols or for likely water droplets). Could the authors comment on the reason for that? Wouldn't the consideration of the Ångström exponents provide confirmation or allow a refinement of the classification?

6. In the explanations in the text about the categorization criteria summarized on table 1, I suggest, for the convenience of the reader, a brief explanation of the Cloudnet algorithm used for some cloud categories.

7. When comparing the results of the lidar categorization to those of a Cloudnet station operating in the same location (or close enough) as the multiwavelength lidar (e.g. lines 467-479, 523-527), the authors conclude that either classification is reasonable, although they do not correspond to the same types of particle, and that the comparison results show the synergy between radar- and lidar-based classification, each system being sensitive to different types of particles. But, can a misclassification be completely ruled out, e.g. the supercooled droplets or the drizzle identified by Cloudnet being mistaken by the large aerosol particles identified by the lidar?

8. On lines 647-649 it is stated: "Ice crystals were also often classified correctly, but sometimes remained unclassified or even false classified as aerosol as a consequence of multiple reasons (a priori information aiming at aerosol, low depolarizing characteristics in certain temperature ranges, etc.)". I think this is not sufficiently emphasize in the paper. These instances where the classification outcome is doubtful should be pointed out in section 4.2

**Other issues**

1. The description given in the text of the $\text{Polly}^{\text{XT}}_{\text{IFT}}$ system does not coincide completely with that found in the reference Engelemann et al. 2016. Only a depolarization channel at 532 nm is mentioned in the text, while two, at 355 nm and 532 nm, are indicated in the reference.

2. On lines 191-192, referring to Fig.1, the authors say that "One can see that during most of the intervals of no setup change, the lidar system parameter is relatively stable and only

some of the setup changes have caused a significant change in $C^\lambda$ ". Is this really sustained by Fig. 1? There seems not to be periods between changes with many $C^\lambda$ measurements.

3.  Also related to Fig.1, I found it difficult to relate the vertical lines indicating changes in the lidar setup to the specific dates mentioned in the text. I suggest labeling those lines (at least those referred to in the text) with the precise dates.

4.  On lines 194-196 the authors say: "It was found that changes in the indoor temperature of the cabinet due air conditioning malfunctioning had led to a change of the alignment and thus a change in $C^\lambda$ during this period". Didn't this also lead to a change of the overlap function?

5.  On lines 199-200 the authors say: "On three days (18 April, 25 April, and 10 May), for which multiple system setup changes were performed, more than one lidar system parameter was used to account for these setup changes". Is this shown in Fig. 1?

6.  In fig.1, is it only an "optical effect", because the absolute values are higher, or it is true that the lidar system parameter has more relative variability for the 1064-nm channel than for the two other channels. If it is true, is the reason known?

7.  On lines 211-212 it is stated: "For days with inappropriate weather conditions a standard value (mean of HOPE) [of the depolarization calibration constant] is used". I suppose this refers to inappropriate weather conditions for determining $V^*$. This should be explained.

8.  On line 262 a molecular depolarization ratio equal to 0.0053 is mentioned for Polly$^{\text{XT}}$. A reference is in order.

9.  The sentence on lines 368-371: "This threshold yields a ratio of molecular to particle backscattering at 532 (355) nm higher than 60 (180) at sea level and thus is valid for a Rayleigh calibration by means of the Raman or Klett-Fernald lidar method which might be one future application of the target categorization presented herein" is not clear (because of its ending: "which might be one future application of the target categorization presented herein") . Please check and clarify.

10.  While the figures have high resolution, which allows to zoom in when reading the paper on a computer, the size of some of them is too small for a comfortable reading in print. This affects especially figs. 3, 4, 7, 8, 9, 10, and 12. In addition the left and central panels of fig. 4 have probably an excess of information, with some curves with similar colors, which makes difficult to distinguish the different parameters represented. A similar problem occurs with fig. 5. I suggest either splitting the figures to reduce the information content of each or using, in addition to colors, different line types to make easier to distinguish between them.

11.  The lidar categorization figures bear the title "Lidar target classification" in an insert, while for the Cloudnet ones the titles read just "Target classification". I suggest that these are modified to read "Cloudnet target classification".

12.  At the beginning of section 4.2 I suggest a brief rationale on the selection of three case studies selected. Why these cases instead of others? What makes them especially interesting?

13.  The convective cloud observed "shortly past 12 UTC" (lines 432 and 458-459) on figs. 7 and 8 is very difficult to distinguish. I suggest some means (arrow, circle around…) to draw the reader's attention to it.

14.  Do the statement on lines 599-601 ("From these pixels, clean (i.e. molecular scattering dominating) atmosphere was observed in 29%, clouds in only 7%, aerosol in about 37% and non-typed particles/particles with low concentration in 27% of the analysed and feature-classified pixels"), and the statistics presented in fig. 13, make sense without specifying the maximum exploration height?

**Minor issues:**

1.  The authors use in many instances the E-notation (e.g. 2e-5 to represent $2 \times 10^{-5}$). The 10 with superscript exponent notation should be used throughout.

2.  There are several instances of "even so" that should probably be "even though".

3.  The paper should be revised to correct typos (e.g "Ångstöm" instead of "Ångström" on line 283, small punctuation issues (e.g. missing commas), missing spaces between words (e.g. line 269 "2013started"), etc.

---

## Referee Comment (RC2) · Anonymous Referee #1 · 21 Mar 2017

Please note that the comments in the posted review apply to the original version of the manuscript (including the line numbering). Some of the remarks have already been taken into account, in particular those referring to the small size of some figures.

---

## Referee Comment (RC3) · Anonymous Referee #2 · 28 Apr 2017

This paper proposes a method on target categorization of aerosol and clouds using calibrated lidar signals (i.e., attenuated backscatter coefficients) at 532 and 1064nm with depolarization ratio at 532nm and demonstrates the performance using ground-based multi-wavelength polarization lidar data. This paper is well written and it is easy for readers to read understand the contents of this paper. The proposed method is based on a commonly-used threshold method, however, a new idea that uses quasi backscatter coefficient and particle depolarization ratio is introduced. The target categorization products evaluated by the developed method are useful for understanding the distribution of aerosol and clouds and their occurrence. The content of this paper is suitable to this journal (AMT). Only comments on minor revision are given as follows:

1) P6. Line 174-180, "For that reason ~~~ Baars et al. 2016". What does the term "hybrid approach" mean? Do you use particle backscatter coefficients derived from Raman lidar measurements for nighttime data and derived by Klett-Fernald method for daytime data to evaluate particle extinction coefficients by multiplying the assumed lidar ratio of 55sr? Readers needs more explanation to understand this part.

2) Figure 1. It seems that the variation of the 1064nm lidar system parameter is larger than the calibration constants at 355nm and 532nm. What is the cause on this larger variation? Regarding to question 1), when you derive the 1064nm lidar system parameter, how do you evaluate the backscatter coefficients at 1064nm? If you use Klett-Fernald method, how you assume the boundary condition (can you find aerosol free layer for the 1064nm data) ?

3) Figure 5 It may be difficult to distinguish each line by difference of only color. It would be better to use solid, dashed, and dotted lines with color difference.

4) Figure 6 "Aerosol typing" is connected with "untyped aerosol/low concentration" by line.

5) p20 Line483, "Therefore, we conclude ~~~simultaneously". It is difficult to "conclude" because there is no evidence to prove that ice and supercooled drops, and large, spherical aerosols coexisted though the lidar and radar measurements indicate the possibility of their co-exsistence as you suggest.

6) P21 Line 526 "identifies large aerosol ~~~ evaporation" The target categorization of CloudNet and the lidar derived target categorization seem to indicate the coexistence of drizzle particle and large, spherical aerosol particles (evapolated drizzle particle) in the area, however, one can suggest that this lidar derived target categorization fails and identifies drizzle (or rain) particles as aerosol particles though you commented in this paper that the categorization of drizzle or rain was beyond scope. I recommend you to mention (or discuss) about possibility of identification (categorization) of drizzle particles using lidar data to make clear the performance and limitation of this target

categorization method.

---

## Author Comment (AC1) · 29 May 2017

Reply to referees:

First of all, we would like to thank the referees for their efforts. The comments and questions have been very helpful and will certainly improve the manuscript.

Find below the Reviewer comments, followed by **the author's response in bold.** Whenever needed and suitable "*changes in the Manuscript are explicitly shown below the response in italic*"

**Most Figures have been redesigned for a better readability without changing the content!**

**Referee 1:**

Comment:
Please note that the comments in the posted review apply to the original version of the manuscript (including the line numbering). Some of the remarks have already been taken into account, in particular those referring to the small size of some figures.

**Main questions:**

1. One of the parameters used in the proposed categorization method is the so-called quasi-particle backscatter coefficients. Profiles of this parameter are obtained assuming a typical lidar ratio and correcting the attenuated backscatter for the molecular extinction and the particle extinction, the latter estimated by multiplying the lidar ratio by a "first guess" of the quasi-particle backscatter coefficient obtained from the attenuated backscatter corrected only for the molecular extinction. Eqs. (6)-(8) describe clearly the procedure. However one wonders if the cycle could not go on – i.e. if the quasi-particle backscatter coefficient so obtained could not be used to refine the estimate of the particle extinction (Eq. (7)), which in turn would be inserted again in Eq. (8) to obtain a new estimate of the quasi-particle backscatter – until a convergence criterion is attained. It doesn't seem that implementing this iterative cycle would be too cumbersome from the computational point of view and it would probably converge to what the authors call the "real backscatter coefficient" on lines 293, 305, and 616, and in the captions of figs. 4 and 5. Have the authors tested if this would improve the performance of the categorization?

**We have considered (and even tested) this approach, but the procedure is similar to the Klett-Fernald forward integration (so in our case with a reference value very close to the surface). I.e., the procedure can be easily numerically unstable as it is strongly dependent on the input lidar ratio. This means, the solution does not necessarily converge if the input lidar ratio is slightly higher than the "real one", which leads to unrealistically high extinction profiles and thus non-reliable backscatter information (much too high…"exploding with height").
On the other hand, taking a lidar ratio always too low for all types of scatterers (e.g. 18 sr – lidar ratio of liquid drops) leads to a dramatically underestimated extinction, which also leads to a non-reliable backscatter information.
Therefore, we consider the approach of using the quasi extinction coefficient as the best estimate for the extinction as it is least sensitive to the a-priori lidar ratio assumption. However, we are aware that this approach leads in many cases to a slightly underestimated extinction. Nevertheless, as the extinction profile is used for**

**transmission correction only, we think our best estimate approach is well suited for our methodology as discussed by Fig. 5 and 6 (former 4 and 5).**

**We added a few lines in the manuscript, where we state why we did not use the iterative procedure, to make this clear.**

*"An iterative approach for the determination of the particle extinction coefficient using the formulas above is not possible, because the solutions do not converge if the input lidar ratio is not exactly identical to the lidar ratio valid for the observed scatterers. If the input lidar ratio is higher than the atmospheric one, the extinction coefficient and thus also the backscatter coefficient is in general overestimated and the procedure quickly approaches unstable solutions. On the other hand, if the lidar ratio input is too low, too small values not increasing during the procedure are obtained. This behavior is similar to the so-called Klett-Fernald Forward Iteration (Klett, 1981; Fernald, 1984), which also relies on a-priori information of the lidar ratio and can be numerically unstable."*

2. I find the term "real backscatter coefficient", as used where indicated in the above remark, somewhat misleading, because, even though the authors do not indicate explicitly how it is obtained (Klett-Fernal algorithm, iterative algorithm?), it relies probably on an assumed lidar ratio. The authors should clarify this.

**You're right, the statement is misleading, we rephrased it at all instances. We added "by Raman or Klett method". I.e., we consider the Raman or Klett solutions as the 'truth' with respect to the estimated "quasi" parameters.**

3. In connection with the previous remark, the authors should also clarify what is understood by the "truth" on line 313.

**We have rephrased it in the same way as for the previous comment.**

4. On lines 14-16, it is stated: "By analyzing the entire HOPE campaign, almost 1 million pixel (5 min times 30 m) could be successfully classified from the two months data set with the newly developed tool". I wonder if the claim (repeated in a somewhat different and possibly less strong and more appropriate way on line 598-599: "more than 1 million pixels of 30 m vertical and 5 min temporal resolution were successfully analysed") is not too exaggerated. If we understand by "successfully classified" that the pixels actually contained the particle class assigned by the algorithm, can the authors be sure of that?

**The statement was misleading, we rephrased it, and now just write "analysed".**

5. In the criteria for the categorization, clearly summarized in table 1, the Ångström exponents are not always taken into account (e.g. for the aerosol mixture, partly non-spherical, for large, non-spherical aerosols or for likely water droplets). Could the authors comment on the reason for that? Wouldn't the consideration of the Ångström exponents provide confirmation or allow a refinement of the classification?

**As the Ångström exponent is not always available, we cannot take it always into account. Therefore, "the likely" categories were introduced for the clouds to give information which are likely but not confirmed by this parameter. For the aerosol classes, we make the distinction on spherical particles primary with the Ångström**

**exponent, but for non-spherical particles the dominant parameter is the particle depolarization ratio, therefore the use of the Ångström exponent is not necessary.**

6. In the explanations in the text about the categorization criteria summarized on table 1, I suggest, for the convenience of the reader, a brief explanation of the Cloudnet algorithm used for some cloud categories.

**This is right and this is already done. See lines 346ff of the original discussion paper you are referring to (page 15, line 5 ff in the online version).**

7. When comparing the results of the lidar categorization to those of a Cloudnet station operating in the same location (or close enough) as the multiwavelength lidar (e.g. lines 467-479, 523-527), the authors conclude that either classification is reasonable, although they do not correspond to the same types of particle, and that the comparison results show the synergy between radar- and lidar-based classification, each system being sensitive to different types of particles. But, can a misclassification be completely ruled out, e.g. the supercooled droplets or the drizzle identified by Cloudnet being mistaken by the large aerosol particles identified by the lidar?

**Right, we cannot completely rule out a misclassification. Thus, we state now that a coexistence of this type of scattereres is in principle possible and very likely in this case. For the discussion of this topic, we now present theoretical evidence combined with observed values. Therefore, we have added a completely new discussion for the 4 April case and added a new Figure showing the simulations:**

*"This example shows the different sensitivity concerning particle size and thus the potential synergy between the lidar- and radar-based classifications. While the lidar is more sensitive to the numerous but comparably small aerosol particles, the radar is most sensitive to the few but large precipitation particles. If we assume a Marshall-Palmer rain droplet number size distribution (Marshall and Palmer, 1948), we can estimate the light extinction of the drizzle in dependence of the rain rate as shown in Fig. 11. For low rain rates, which have occurred in the case of 4 April 2013 because no precipitation reached the ground, extinction coefficients well below typical aerosol values are calculated. Aerosol extinction in the PBL was about 150 to 200 Mm^-1 throughout the observation time in the case presented here. At a height of 1.5 km, which is 250 m below the cloud base, extinction coefficients of about 100 Mm^-1 were observed at 4 UTC. When no clouds were present at 1 UTC, they were 35 to 50 Mm^-1 at this height. Thus, if one considers hygroscopic growth, one can conclude that the lidar signal was dominated by aerosol instead of the few drizzle droplets even though they also contributed to the lidar return. On the other hand, as the radar is sensitive to the sixth power of the diameter of the scatterers (while the lidar is to the power of 2), it is sensitive to the few but large precipitation droplets. Therefore, the Cloudnet classification defines the region of interest to contain ice and supercooled drops and ice only - putting the priority on the cloud-sensitive radar observations. Given the added value of the multiwavelength lidar aerosol classification, we can however conclude that between 3 and 10 UTC all detected features, i.e., large, spherical aerosol particles and ice and supercooled drops were present simultaneously, even though the full instrument synergy of the here presented instruments is still a current research topic. "*

New Figure:

[Figure]

8. On lines 647-649 it is stated: "Ice crystals were also often classified correctly, but sometimes remained unclassified or even false classified as aerosol as a consequence of multiple reasons (a priori information aiming at aerosol, low depolarizing characteristics in certain temperature ranges, etc.)". I think this is not sufficiently emphasize in the paper. These instances where the classification outcome is doubtful should be pointed out in section 4.2

**We have discussed this issue now intensively in the end of Section 4.2.2:**

"*However, as can be seen as well in Fig. 10a, ice crystals are often classified correctly, but sometimes remain unclassified or are even false classified as aerosol. The reason for the non-classification of ice crystals is mostly the lack of depolarization information at 532 nm while the 1064 nm channel is able to detect particles especially at high altitudes at which the SNR of the 532 nm channels is too low. This occurs e.g. for the thin ice cloud at about 10 km past 21:30 UTC. The frequency of occurrence of misclassification of ice crystals as aerosol is increasing with increasing penetration depth of the ice clouds as can be seen in Fig. 10a past 16 UTC in the height range of 4 – 7 km. The reason for that false classification is the used a priori information aiming on aerosol (i.e. the lidar ratio and Ångström exponent). This leads to a wrong attenuation correction and thus to wrong quasi particle backscatter coefficient and quasi particle depolarization ratio values above the cloud base. Furthermore, multiple scattering at the large cloud hydrometeors leads to an additional underestimation of the light attenuation (see, e.g., Seifert et al. 2007, Kienast-Sjögren et al. 2016, or Gouveia et al. 2017). For that reason, the current lidar-standalone approach is trustworthy only at cloud bases and a few tens of meters above depending on the cloud optical thickness. Nevertheless, the pixels above an ice cloud base are shown as it might be of interest for research and false classification are comparable low with respect to correct classifications as also seen in Fig. 10a. As explained in the outlook, it is planned to combine the current approach with the Cloudnet one and we think that this shortcoming can be overcome when the use of cloud radar information allows to set other a priori information for clouds than for aerosol.*

**Other issues**

1. The description given in the text of the Polly$^{\text{XT}}_{\text{IFT}}$ system does not coincide completely with that found in the reference Engelemann et al. 2016. Only a depolarization channel at 532 nm is mentioned in the text, while two, at 355 nm and 532 nm, are indicated in the reference.

**In Engelmann et al., 2016 it is correctly stated that PollyXT_IfT has only one depolarization channel at 532 nm as also written in our manuscript.**
**In fact, this system was in the beginning equipped with a 355-nm depolarization channel, but this was changed in 2011, when depolarization at 355 nm was removed and a 407 water vapour channel and depolarization at 532 nm was installed.**

2. On lines 191-192, referring to Fig.1, the authors say that "One can see that during most of the intervals of no setup change, the lidar system parameter is relatively stable and only some of the setup changes have caused a significant change in $C^\lambda$ ". Is this really sustained by Fig. 1? There seems not to be periods between changes with many $C^\lambda$ measurements.

**We have rephrased the corresponding statement and now only claim that the lidar system parameter is relatively stable without referring to the maintenance intervals. As the daily mean values are shown it indeed appears to be that there are not too many measurements in between. But also the daily evolution of the lidar system parameters shows that if no changes in the setup took place, the system parameter did not change significantly.**

3. Also related to Fig.1, I found it difficult to relate the vertical lines indicating changes in the lidar setup to the specific dates mentioned in the text. I suggest labeling those lines (at least those referred to in the text) with the precise dates.

**We have redesigned Figure 1 according to your comments. Furthermore, we have realized that the time of day was not considered in the previous plot version. We have changed that now.**

4. On lines 194-196 the authors say: "It was found that changes in the indoor temperature of the cabinet due air conditioning malfunctioning had led to a change of the alignment and thus a change in $C^\lambda$ during this period". Didn't this also lead to a change of the overlap function?

**Yes, it is affected and this is also what we mean with "alignment". We made this clear in the text now.**

*"...had led to a change of the alignment (e.g. the overlap between the receiver field of view and the laser beam) and thus a change in C during the period."*

5. On lines 199-200 the authors say: "On three days (18 April, 25 April, and 10 May), for which multiple system setup changes were performed, more than one lidar system parameter was used to account for these setup changes". Is this shown in Fig. 1?

**Now, with the redesigned Figure 1 it is shown. Before, time of day was not considered and thus different system parameter at one day were difficult to identify.**

6.  In fig.1, is it only an "optical effect", because the absolute values are higher, or it is true that the lidar system parameter has more relative variability for the 1064-nm channel than for the two other channels. If it is true, is the reason known?

**It's an "optical effect". The relative changes are in the same order of magnitude. This is noted now explicitly in the text!**

*"The relative change of the lidar system parameter is similar for all three wavelengths, even though it looks different in Fig. 1 due to the scaling applied."*

7.  On lines 211-212 it is stated: "For days with inappropriate weather conditions a standard value (mean of HOPE) [of the depolarization calibration constant] is used". I suppose this refers to inappropriate weather conditions for determining $V*$. This should be explained.

**Thanks for the advice. We have made this clear now in the text!**

8.  On line 262 a molecular depolarization ratio equal to 0.0053 is mentioned for Polly$^{XT}$. A reference is in order.

**Done! We added the reference of Behrendt and Nakamura (2002).**

9.  The sentence on lines 368-371: "This threshold yields a ratio of molecular to particle backscattering at 532 (355) nm higher than 60 (180) at sea level and thus is valid for a Rayleigh calibration by means of the Raman or Klett-Fernald lidar method which might be one future application of the target categorization presented herein" is not clear (because of its ending: "which might be one future application of the target categorization presented herein") . Please check and clarify.

**We have clarified this in the text now.**

*".. and thus is valid for a Rayleigh calibration by means of the Raman or Klett-Fernald lidar method. One future application of the target categorization presented herein might be to find appropriate regions for Rayleigh calibration, i.e. height regions with almost pure molecular scattering but sufficient high SNR."*

10.  While the figures have high resolution, which allows to zoom in when reading the paper on a computer, the size of some of them is too small for a comfortable reading in print. This affects especially figs. 3, 4, 7, 8, 9, 10, and 12. In addition the left and central panels of fig. 4 have probably an excess of information, with some curves with similar colors, which makes difficult to distinguish the different parameters represented. A similar problem occurs with fig. 5. I suggest either splitting the figures to reduce the information content of each or using, in addition to colors, different line types to make easier to distinguish between them.

**This comment is related to the first submitted version and according to the reviewers comment, we have already redesigned all the graphs before publishing in AMTD.**

11.  The lidar categorization figures bear the title "Lidar target classification" in an insert, while for the Cloudnet ones the titles read just "Target classification". I suggest that these are modified to read "Cloudnet target classification".

**Thanks for the suggestion. While redesigning all Figures, we have also included this suggestion.**

12. At the beginning of section 4.2 I suggest a brief rationale on the selection of three case studies selected. Why these cases instead of others? What makes them especially interesting?

**Done!**

*"These examples cases represent a wide variety of different meteorological situations and are therefore well suited to demonstrate the capabilities of the newly developed lidar target categorization."*

13. The convective cloud observed "shortly past 12 UTC" (lines 432 and 458-459) on figs.7 and 8 is very difficult to distinguish. I suggest some means (arrow, circle around…) to draw the reader's attention to it.

**Thanks for the advise, we have done so!**

14. Do the statement on lines 599-601 ("From these pixels, clean (i.e. molecular scattering dominating) atmosphere was observed in 29%, clouds in only 7%, aerosol in about 37% and non-typed particles/particles with low concentration in 27% of the analysed and feature-classified pixels"), and the statistics presented in fig. 13, make sense without specifying the maximum exploration height?

**Before this statement, we have clearly discussed the shortcoming of the target categorization and also have stated that we cannot always observe the whole troposphere (see some lines above:"… We have to repeat that this cloud statistic is biased as the lidar can penetrate liquid clouds only by a few tens of meters. "). However, we think that it still makes sense to state what kind of scatterers could be analysed how often.**

Minor issues:

1. The authors use in many instances the E-notation (e.g. 2e-5 to represent $2 \times 10^{-5}$ ). The 10 with superscript exponent notation should be used throughout.

**We think this kind of numbering is correct and it also helps to improve the readability of the low numbers. However, we have homogenised our numbering throughout the whole manuscript as we agree that a mixed notation is not favorable.**

2. There are several instances of "even so" that should probably be "even though".

**Done!**

3. The paper should be revised to correct typos (e.g "Ångstöm" instead of "Ångström" on line 283, small punctuation issues (e.g. missing commas), missing spaces between words (e.g. line 269 "2013started"), etc.

**Done!**

---

## Author Comment (AC2) · 29 May 2017

Reply to referees:

First of all, we would like to thank the referees for their efforts. The comments and questions have been very helpful and will certainly improve the manuscript.

Find below the Reviewer comments, followed by **the author's response in bold.** Whenever needed and suitable *"changes in the Manuscript are explicitly shown below the response in italic"*

**Most Figures have been redesigned for a better readability without changing the content!**

**Referee 2:**

1) P6. Line 174-180, "For that reason ~~~ Baars et al. 2016". What does the term "hybrid approach" mean?

**As non-native speakers, we probably misused the term "hybrid". We have changed the comment accordingly (see below).**

- Do you use particle backscatter coefficients derived from Raman lidar measurements for nighttime data and derived by Klett-Fernald method for daytime data to evaluate particle extinction coefficients by multiplying the assumed lidar ratio of 55sr?

**Yes, this is exactly what we mean. We have made this clear in the text, see new text below.**

- Readers needs more explanation to understand this part.

**Thanks for the suggestions. We rephrased the paragraph to make it more clear.**

*"For that reason and because the particle extinction coefficient derived with the Raman method is only available during night time, we introduced a **2-step approach** to estimate the particulate transmission needed to solve Eq. 2. First, we calculate the particle extinction coefficient profile derived from the particle backscatter coefficient profile (**Raman or Klett - depending on time of day**) multiplied with a constant lidar ratio of 55 sr as a good compromise of the lidar ratio values observed during HOPE and at other European continental sites...."*

2) Figure 1. It seems that the variation of the 1064nm lidar system parameter is larger than the calibration constants at 355nm and 532nm. What is the cause on this larger variation?

**In fact, the relative variation is the same, but it looks stronger in Fig.1 due to the scaling....we add a sentence in the text to clarify that!**

*"The relative change of the lidar system parameter is similar for all three wavelengths, even though it looks different in Fig. 1 due to the scaling applied."*

5) Regarding to question 1), when you derive the 1064nm lidar system parameter, how do you evaluate the backscatter coefficients at 1064nm? If you use Klett- Fernald method, how you assume the boundary condition (can you find aerosol free layer for the 1064nm data) ?

**The backscatter coefficients are determined as described in Baars, ACP, 2016,"PollyNET - …" as stated in the manuscript: We use a Rayleigh fit procedure to obtain atmospheric regions of almost pure molecular scattering which are then used as reference height. Then, the backscatter coefficient is calculated with a reference value of 1e-6 km^-1 sr^-1 with either the Klett-Fernald method or the Raman method (using 607 nm signal) depending on the SNR in the 607 nm channel. In Polly systems, photon counting is used to detect the 1064 nm signal, which allows us in most cases to detect the weak molecular contribution at this wavelength. However, this is not always possible and as result a slightly lower number of backscatter profiles at 1064 nm compared to 532 and 355 nm are obtained.**

3) Figure 5 It may be difficult to distinguish each line by difference of only color. It would be better to use solid, dashed, and dotted lines with color difference.

**We have redesigned this Figure according to your suggestions and furthermore have applied a 5-bin vertical smoothing which significantly increased the readability.**

4) Figure 6 "Aerosol typing" is connected with "untyped aerosol/low concentration" by line.

**You are right, this is confusing. We have corrected this.**

5) p20 Line483, "Therefore, we conclude ~~~simultaneously". It is difficult to "con- clude" because there is no evidence to prove that ice and supercooled drops, and large, spherical aerosols coexisted though the lidar and radar measurements indicate the possibility of their co-exsistence as you suggest.

**See response below.**

6) P21 Line 526 "identifies large aerosol ~~~ evaporation" The target categorization of CloudNet and the lidar derived target categorization seem to indicate the coexistence of drizzle particle and large, spherical aerosol particles (evapolated drizzle particle) in the area, however, one can suggest that this lidar derived target categorization fails and identifies drizzle (or rain) particles as aerosol particles though you commented in this paper that the categorization of drizzle or rain was beyond scope. I recommend you to mention (or discuss) about possibility of identification (categorization) of drizzle particles using lidar data to make clear the performance and limitation of this target categorization method.

**As also requested by the first referee, we state now that a coexistence of this type of scattereres is in principle possible and very likely in this case. Therefore, we present theoretical evidence combined with observed values. To make this topic clear, we have added a completely new discussion for the 4 April case and added a new Figure showing the simulations:**

*"This example shows the different sensitivity concerning particle size and thus the potential synergy between the lidar- and radar-based classifications. While the lidar is more sensitive to the numerous but comparably small aerosol particles, the radar is most sensitive to the few but large precipitation particles. If we assume a Marshall-Palmer rain droplet number size distribution (Marshall and Palmer, 1948), we can estimate the light extinction of the drizzle in dependence of the rain rate as shown in Fig. 11. For low rain rates, which have occurred in the case of 4 April 2013 because no precipitation reached the ground, extinction coefficients*

*well below typical aerosol values are calculated. Aerosol extinction in the PBL was about 150 to 200 Mm^-1 throughout the observation time in the case presented here. At a height of 1.5 km, which is 250 m below the cloud base, extinction coefficients of about 100 Mm^-1 were observed at 4 UTC. When no clouds were present at 1 UTC, they were 35 to 50 Mm^-1 at this height. Thus, if one considers hygroscopic growth, one can conclude that the lidar signal was dominated by aerosol instead of the few drizzle droplets even though they also contributed to the lidar return. On the other hand, as the radar is sensitive to the sixth power of the diameter of the scatterers (while the lidar is to the power of 2), it is sensitive to the few but large precipitation droplets. Therefore, the Cloudnet classification defines the region of interest to contain ice and supercooled drops and ice only - putting the priority on the cloud-sensitive radar observations. Given the added value of the multiwavelength lidar aerosol classification, we can however conclude that between 3 and 10 UTC all detected features, i.e., large, spherical aerosol particles and ice and supercooled drops were present simultaneously, even though the full instrument synergy of the here presented instruments is still a current research topic."*

New Figure: